# An ensemble Kalman filter system with the Stony Brook Parallel Ocean Model v1.0

Shun Ohishi[1,2,3], Tsutomu Hihara[4], Hidenori Aiki[3,5], Joji Ishizaka[3], Yasumasa Miyazawa[5], Misako Kachi[6], Takemasa Miyoshi[1,2,7]

[1]RIKEN Center for Computational Science, Kobe, 6500047, Japan
[2]RIKEN Cluster for Pioneering Research, Kobe, 6500047, Japan
[3]Institute for Space-Earth Environmental Research, Nagoya University, Nagoya, 4648601, Japan
[4]Japan Fisheries Information Service Center, Tokyo, 1040055, Japan
[5]Application Laboratory, Japan Agency for Marine-Earth Science and Technology, Yokohama, 2360001, Japan
[6]Earth Observation Research Center, Japan Aerospace Exploration Agency, Tsukuba, 3058505, Japan
[7]RIKEN Interdisciplinary Theoretical and Mathematical Sciences Program (iTHEMS), Kobe, 6500047, Japan

*Correspondence to*: Shun Ohishi (shun.ohishi@riken.jp) and Takemasa Miyoshi (takemasa.miyoshi@riken.jp)

**Abstract.** This study develops an ensemble Kalman filter (EnKF)-based regional ocean data assimilation system, in which the local ensemble transform Kalman filter (LETKF) is implemented with the Stony Brook Parallel Ocean Model (sbPOM) version 1.0 to assimilate satellite and in-situ observations at a daily frequency. A series of sensitivity experiments are performed with various settings of the incremental analysis update (IAU) and covariance inflation methods, for which the relaxation-to-prior perturbations and spread (RTPP and RTPS, respectively) and multiplicative inflation (MULT) are considered. We evaluate the geostrophic balance and the analysis accuracy compared with the control experiment in which the IAU and covariance inflation are not applied. The results show that the IAU improves the geostrophic balance, degrades the accuracy, and reduces the ensemble spread, and that the RTPP and RTPS have the opposite effect. The experiment using the combination of the IAU and RTPP results in significant improvement for both balance and analysis accuracy when the RTPP parameter is 0.8–0.9. The combination of the IAU and RTPS improves the balance when the RTPS parameter is ≤0.8 and increases the analysis accuracy for the parameter values between 1.0 and 1.1, but the balance and analysis accuracy are not improved significantly at the same time. The experiments with MULT inflating forecast ensemble spread by 5 % do not demonstrate sufficient skill in maintaining the balance and reproducing the surface flow field regardless of whether the IAU is applied or not. 11-day-long ensemble forecast experiments show consistent results. Therefore, the combination of the IAU and RTPP with the parameter of 0.8–0.9 is found to be the best setting for the EnKF-based ocean data assimilation system.

**Short summary (484 characters).** We develop an ensemble Kalman filter-based regional ocean data assimilation system, in which satellite and in-situ observations are assimilated at a daily frequency. We find the best setting for dynamical balance and accuracy based on sensitivity experiments on how to inflate the ensemble spread and how to apply the analysis update to the model evolution. This study has a broader impact on more general data assimilation systems in which the initial shocks are a significant issue.

## 1. Introduction

The ensemble Kalman filter (EnKF; Evensen, 1994, 2003) estimates optimal analyses using model forecasts and observations with their error covariance. The EnKF has advantages in including flow-dependent forecast errors from an ensemble of model forecasts and in its relative ease of implementation with various models. Therefore, various EnKF-based ocean data assimilation systems have been developed thus far (cf. Table 1).

The number of observations has increased dramatically with enhanced observations of temperature and salinity in the ocean interior by Argo profiling floats, and measurements of sea surface temperature, salinity, and height (SST, SSS, and SSH, respectively) by satellites. The geostationary satellite Himawari-8 (Bessho et al., 2016; Kurihara et al., 2016) has an infrared sensor observing SSTs in the Pacific region since July 2015, although there are missing values where cloud obscures the sea surface. The geostationary orbit and short observation interval allow Himawari-8 to provide better daily coverage within the observation area than a polar orbiting satellite with a microwave sensor, such as the Global Change Observation Mission-Water (GCOM-W; https://gportal.jaxa.jp/gpr), which can capture SSTs even in cloudy regions. Satellite SSS observations by Soil Moisture and Ocean Salinity (SMOS) started in June 2010, and previous studies demonstrated their positive impacts with their ocean data assimilation systems to better represent the ocean interior structure such as mixed and barrier layers, low salinity water caused by river discharge, and prediction of the El Niño-Southern Oscillation (ENSO; Chakraborty et al., 2014; Hackert et al., 2014; Toyoda et al., 2015). The Surface Water and Ocean Topography (SWOT; https://swot.jpl.nasa.gov/) satellite that has a new type of altimeter observing SSH anomalies (SSHAs) in two dimensions over a 120 km wide swath is scheduled for launch in 2022.

To take advantage of such enhanced observations, frequent data assimilation is important. Here, dynamical imbalances in the analysis field may cause an initial shock with high-frequency gravity waves and may degrade the analysis accuracy. He et al. (2020) described the relationship between the assimilation interval and accuracy using an atmospheric data assimilation system. As seen in Table 1, most of the recent ocean data assimilation systems have an assimilation interval longer than 5 days: in particular, 5- and 7-day assimilation intervals are employed in the existing ocean reanalysis datasets of the Predictive ocean atmosphere model for the Australia Ensemble Ocean Data Assimilation System (PEODAS; Yin et al.,

2011) and TOPAZ4 (Sakov et al., 2012), respectively. The PEODAS assimilates only in-situ temperature and salinity data, whereas the TOPAZ4 uses all types of observations but with inflation of observation errors. Although ocean data assimilation systems constructed by Karspeck et al. (2013) and Miyazawa et al. (2012) have short assimilation intervals of 1 and 2 days, respectively, the former assimilates only in-situ temperature and salinity data, and the latter conducts a data assimilation experiment for a short period of 20 days because unrealistic fields are detected if the experiment is performed over several months (Y. Miyazawa 2022 personal communications). Although Brüning et al. (2021) recently established regional data assimilation systems for the North Sea and Baltic Sea at a frequent interval of 12 hours, only satellite SSTs are assimilated. Therefore, the existing systems might mitigate the effects of the initial shock by using the longer assimilation interval, inflating observation errors, and reducing the number of assimilated observations. This is also the case for atmosphere-ocean coupled data assimilation systems (e.g. Brune et al., 2015; Chang et al., 2013; Counillon et al., 2016; Tang et al., 2020). To provide accurate analyses in EnKF-based ocean data assimilation system in which satellite and in-situ observations are assimilated at a frequent interval, it is necessary to investigate an optimal setting for both dynamical balance and accuracy.

The incremental analysis update (IAU; Bloom et al., 1996; see Sect. 2.1) has been proposed to reduce noise from high-frequency gravity waves associated with the initial shock. Covariance relaxation methods such as relaxation-to-prior perturbations and spread [RTPP (Zhang et al., 2004) and RTPS (Whitaker and Hamill, 2012), respectively; see Sect. 2.3], in which the analysis ensemble perturbations are relaxed towards the forecast ensemble perturbations, would also mitigate the initial shock (Houtekamer and Zhang, 2016; Ying and Zhang, 2015). In EnKF-based ocean data assimilation system, how to apply the analysis update to the model evolution and how to inflate the ensemble spread could make significant differences for the dynamical balance and accuracy. However, the IAU and RTPP/RTPS have not been widely used in an EnKF-based ocean data assimilation systems (Table 1). Therefore, this study aims to develop an EnKF-based ocean data assimilation system with a frequent assimilation interval of 1 day for taking advantage of frequent satellite observations, and to explore the optimal settings by performing sensitivity experiments with various settings of the IAU and covariance inflation methods.

This paper is organized as follows. Section 2 describes data and methods about IAU, RTPP, and other schemes and how to evaluate geostrophic balance and accuracy relative to observations. The details of the EnKF-based ocean data assimilation system and sensitivity experiment are described in Sect. 3. Section 4 presents the results for geostrophic balance and accuracy in the sensitivity experiments. Section 5 compares the prescribed multiplicative inflation (MULT) parameter with the sensitivity experiment with RTPP and IAU. Finally, Sect. 6 provides a summary.

## 2. Data and Methods

In this section, we provide details of the methods used to alleviate some of the problems associated with high frequency assimilation. Section 2.1 presents the IAU designed to cut off noise from high-frequency gravity waves, and Sect. 2.2 describes perturbed boundary conditions. Covariance inflation methods to prevent underestimation of ensemble-based forecast error covariance by various factors such as the limited ensemble size and model imperfections are introduced in Sect. 2.3, and the methods used to evaluate geostrophic balance and accuracy relative to observations in Sect. 2.4.

## 2.1. IAU

In this study, we implement the IAU (Bloom et al., 1996) based on existing ocean data assimilation systems (Balmaseda et al., 2015; Martin et al., 2015). The procedure for one assimilation cycle is as follows: (i) conduct model integration up to the middle of an assimilation window; (ii) assimilate observations within the window and save the analysis increments in temperature, salinity, and horizontal velocity; and (iii) conduct model integration over the assimilation window adding the increments equally distributed to each timestep. The IAU reduces noise from high-frequency gravity waves associated with the initial shock, but the computational costs for the model integration are 1.5 times those for the standard method where the analyses performed at the beginning of the window are used for the model initial conditions. Following Miyazawa et al. (2012), all analysis variables (SSH, temperature, salinity, and horizontal velocities) are used for initial conditions in the standard method. Although there are various IAU methods, the SSH increments are not included in most of the existing ocean data assimilation systems (table 2 of Martin et al., 2015) mainly because the SSH increments tend to cause initial shocks. Even without the SSH increments, the SSH would be modified properly in response to the temperature and salinity increments. Therefore, we adopt the analysis increments of temperature, salinity, and horizontal velocity except for SSH.

## 2.2. Perturbed boundary conditions

Following previous studies (Kunii and Miyoshi, 2012; Penny et al., 2013; Torn et al., 2006), atmospheric and lateral boundary conditions are artificially perturbed for each ensemble member. Atmospheric forcing of the $i$th ensemble member at a time $t$, $\boldsymbol{w}^{(i)}(t)$, is given by

$$\boldsymbol{w}^{(i)}(t) = \boldsymbol{w}(t) + \alpha \left[ \boldsymbol{w}(t + \delta t_i) - \frac{1}{n} \sum_{i=1}^{n} \boldsymbol{w}(t + \delta t_i) \right], \tag{1}$$

where $\boldsymbol{w}(t)$ is atmospheric forcing at a time $t$, $\alpha$ $(= 0.2)$ is an arbitrary constant, $\boldsymbol{w}(t + \delta t_i)$ is atmospheric forcing at the same time as $\boldsymbol{w}(t)$ but in a different year, and $n$ (=100) is the ensemble size. Here, the year in $\boldsymbol{w}(t + \delta t_i)$ is changed every month. As is clear from Eq. (1), the ensemble mean of the atmospheric forcing $\boldsymbol{w}^{(i)}(t)$ is equivalent to $\boldsymbol{w}(t)$.

Lateral boundary conditions for each ensemble member are obtained from a monthly-mean global ocean reanalysis
dataset for different years. Namely, the ensemble mean of the lateral boundary condition corresponds to a monthly
climatology. These perturbed atmospheric and lateral boundary conditions play a role equivalent to additive inflation
(Houtekamer and Zhang, 2016).

### 2.3. Covariance inflation methods

Three covariance inflation methods (MULT, RTPP, and RTPS) are adopted in this study. MULT inflates forecast
error covariance $P^f$ by a factor of $\rho$ ($> 1$):

$$P_{inf}^f = \rho P_{orig}^f, \tag{2}$$

where the subscripts $inf$ and $orig$ denote inflated and original (i.e., before inflation), respectively. Both RTPP and RTPS
restore the analysis ensemble perturbation towards the forecast ensemble perturbations maintaining the analysis ensemble
mean, as represented by

$$X_{inf}^a = \alpha_{RTPP} X^f + (1 - \alpha_{RTPP}) X_{orig}^a \text{ and} \tag{3}$$

$$X_{inf}^{a(i)} = \frac{\alpha_{RTPS} \sigma^{f(i)} + (1 - \alpha_{RTPS}) \sigma_{orig}^{a(i)}}{\sigma_{orig}^{a(i)}} X_{orig}^{a(i)}. \tag{4}$$

Here, $X[= (x^{(1)} - \overline{x}, ..., x^{(i)} - \overline{x}, ..., x^{(n)} - \overline{x})]$ is the ensemble perturbation matrix whose $i$th column consists of the
perturbations of the $i$th ensemble member, where $x^{(i)}$ and $\overline{x}$ are the state vector of the $i$th ensemble member and ensemble
mean, the superscripts $a$ and $f$ denote analysis and forecast, and $\alpha_{RTPP}$ and $\alpha_{RTPS}$ are the relaxation parameters in the RTPP
and RTPS, respectively. $\sigma^{(i)}$ is the ensemble spread of $i$th variable of state vector $x$, as represented by

$$\sigma^{(i)} = \sqrt{(n-1)^{-1} X^{(i)} (X^{(i)})^T}. \tag{5}$$

In the RTPP and RTPS, the relaxation parameters are generally defined between 0 and 1, where $\alpha_{RTPP} = \alpha_{RTPS} = 0$
corresponds to no inflation, and $\alpha_{RTPP} = \alpha_{RTPS} = 1$ corresponds that the analysis ensemble spread is inflated to be

equivalent to the forecast ensemble spread. The RTPP and RTPS are thought to have side effects in maintaining the dynamic balance (Houtekamer and Zhang, 2016; Ying and Zhang, 2015).

## 2.4 Validation

### 2.4.1 Nonlinear balance equation (NBE)

Surface horizontal velocity can be represented as the sum of surface geostrophic and ageostrophic velocities under the geostrophic approximation. Here, the ageostrophic velocity is defined to be caused by the surface wind stress curl except for the vertical geostrophic shear according to the classical Ekman theory (Cronin and Tozuka, 2016). In this study, the atmospheric field is not included in the model state vector, and therefore there are no differences between the forecast and analysis ageostrophic velocities. Consequently, writing the geostrophic balance equation in terms of analysis increments, we obtain

$$f\mathbf{k} \times \delta\mathbf{u} = -g\boldsymbol{\nabla}_h \delta\eta, \tag{6}$$

where $f$ is the vertical component of the Coriolis parameter, $\mathbf{k}$ is a unit vector in the vertical upward direction, $\delta$ is the analysis increment, $\mathbf{u}$ is the horizontal velocity at the sea surface, $g$ $(= 9.8\ m\ s^{-2})$ is the gravitational acceleration, $\boldsymbol{\nabla}_h = (\partial/\partial x, \partial/\partial y)$ is the horizontal gradient operator, and $\eta$ is the SSH. By taking $\partial/\partial x$ of the $x$-component of Eq. (6) plus $\partial/\partial y$ of the $y$-component, Eq. (6) can be reduced to the nonlinear balance equation (NBE; Shibuya et al., 2015; Zhang et al., 2001):

$$-f\delta\zeta + \beta\delta u + g\boldsymbol{\nabla}_h^2 \delta\eta = 0, \tag{7}$$

where $\zeta\,(= \partial v/\partial x - \partial u/\partial y)$ is the relative vorticity at the sea surface and $\beta\,(= \partial f/\partial y)$ is the planetary vorticity gradient. If geostrophic balance is not satisfied in the analysis field, there is an absolute residual of the NBE, $\Delta NBE$:

$$\Delta NBE \equiv \left|-f\delta\zeta + \beta\delta u + g\boldsymbol{\nabla}_h^2 \delta\eta\right|, \tag{8}$$

where $|\cdot|$ denotes taking the absolute value. Smaller (larger) $\Delta NBE$ indicates more (less) geostrophic balance in the analysis field. Few initial shocks would occur if the analysis increments of SSH and surface horizontal velocity satisfy the geostrophic balance.

**2.4.2 Improvement ratio (IR)**

To compare the geostrophic balance and accuracy among sensitivity experiments using a statistical method, we calculate improvement ratios (IRs) of area-averaged $\Delta NBE$ and root mean square deviations (RMSDs) relative to observations as represented by

$$IR_N = \frac{(\Delta NBE)_{CTL} - (\Delta NBE)_{EXP}}{(\Delta NBE)_{CTL}} \times 100 \quad \text{and} \tag{9}$$

$$IR_R = \frac{(RMSD)_{CTL} - (RMSD)_{EXP}}{(RMSD)_{CTL}} \times 100, \tag{10}$$

respectively. The subscripts $CTL$ and $EXP$ indicate control and sensitivity experiments, respectively. Significant improvement and degradation of the dynamical balance and accuracy are detected by applying the bootstrap method, where the IRs of the area-averaged $\Delta NBE$ and RMSDs are resampled for 10,000 cycles and a 99% confidence level is used to detect the significance in all sensitivity assimilation experiments.

**2.4.3 Observations**

To validate the accuracy of a sensitivity experiment, we use observational gridded SSH and SSHA datasets from Archiving Validation and Interpretation of Satellite Oceanographic data (AVISO; Ducet et al., 2000) with horizontal resolution of 0.25°, in-situ surface horizontal velocity from surface drifting buoys of the Global Drifter Program (Elipot et al., 2016), in-situ temperature and salinity in the depth range 1–525 m, and horizontal velocity in the depth range 8–36 m at 144.6°E, 32.3°N south of the Kuroshio Extension (KE) from the Kuroshio Extension Observatory (KEO) buoy (https://www.pmel.noaa.gov/ocs/; see Fig. 5a). The mean dynamical ocean topography (MDOT) of the AVISO is estimated from a geoid model, satellite altimetry, and in-situ drifter buoy data. The AVISO dataset is not an independent observational dataset because satellite SSHAs are used for the assimilation in this study, whereas the surface drifter and KEO buoys are independent. Although validation in the ocean interior might not be sufficient, this is due to the limitation of available independent observations.

**3. EnKF-based ocean data assimilation system**

**3.1. Ocean model**

The sigma-coordinate regional ocean model used in this study is based on the Stony Brook Parallel Ocean Model (sbPOM; Mellor, 2002; Jordi and Wang, 2012) version 1.0, and constructed for the northwestern Pacific region [117°E–180°, 15°–50°N] with horizontal resolution of 0.25° and 50 σ-layers (Table 2). The bottom topography is derived from ETOPO1, a 1 arc-minute global relief model of Earth's surface (Amante and Eakins, 2009). We apply a Gaussian filter with *e*-folding scales of 200 km to the topography to reduce the pressure gradient errors in sigma coordinate models caused by steep bottom slopes (Mellor et al., 1994) and fulfill the condition $|H_{i+1} - H_i|/|H_{i+1} + H_i| < 0.2$, where $H_i$ and $H_{i+1}$ are bottom topographies at adjacent grids. Monthly (Seasonal) temperature and salinity climatologies from World Ocean Atlas 2018 (WOA18; Locarnini et al., 2019; Zweng et al., 2019) with horizontal resolution of 1° and 57 (102) layers are used for an initial condition over depths shallower (deeper) than 1500 m. Lateral boundary conditions for temperature, salinity, and horizontal velocity are obtained from Simple Ocean Data Assimilation (SODA; Carton et al. 2018) version 3.7.2 with horizontal resolution of 0.5° and 50 layers. Here, to satisfy volume conservation, flow relaxation (Guo et al., 2003) is applied to the horizontal velocity at the lateral boundary. The Japanese 55-year Reanalysis (JRA55; Kobayashi et al. 2015) with horizontal and temporal resolution of 1.25° and 6 hours, respectively, is adopted for the atmospheric boundary conditions including air temperature and specific humidity at 2 m, wind velocity at 10 m, shortwave radiation, total cloud fraction, sea level pressure, and precipitation. We also use river discharge from the Japan Aerospace Exploration Agency (JAXA)'s land surface and river simulation system, Today's Earth (TE)-Global (https://www.eorc.jaxa.jp/water/), with horizontal and temporal resolution of 0.25° and 3 hours, respectively. The atmospheric and lateral boundary conditions are perturbed as described in Sect. 2.2, except for the rainfall and river discharge.

The model is driven by wind stresses and heat and freshwater fluxes using bulk formulae in which bulk coefficients are estimated from the Coupled Ocean-Atmosphere Response Experiment (COARE) version 3.5 bulk algorithm (Brodeau et al., 2017; Edson et al., 2013). The horizontal diffusivity coefficient is calculated by a Smagorinsky type formulation with a coefficient of 0.1 (Smagorinsky et al., 1965) and is assumed to be one-fifth of the horizontal viscosity coefficient. The vertical diffusivity coefficient is estimated by the Level 2.5 version of Nakanishi and Niino (2009). The model is spun up from 1 January 2011 to 6 July 2015 using the initial condition with no motion. During the spin-up period, simulated temperatures and salinity are nudged towards the monthly and seasonal climatologies from WOA18 with a 90-day timescale to damp northward overshooting of the Kuroshio. We have confirmed that the perturbed boundary conditions substantially increase the ensemble spread even with the nudging (figure not shown).

### 3.2. Data assimilation

We implement the three-dimensional local ensemble transform Kalman filter (3D-LETKF; Hunt et al. 2007; Miyoshi and Yamane 2007) with 100 ensemble members to assimilate the following observations on a 1 day assimilation interval (Table 3): satellite SSTs from Himawari-8 and GCOM-W, SSS from the SMOS (http://www.esa.int/Applications/Observing_the_Earth/SMOS) and Soil Moisture Active Passive (SMAP) version 4.3 (Meissner et al., 2018), SSH consisting of satellite SSH anomalies from the Copernicus Marine Environment Monitoring Service (CMEMS; http://marine.copernicus.eu/) and MDOT estimated from simulated SSH averaged in 2012–14, and in-situ temperature and salinity from the Global Temperature and Salinity Profile Programme (GTSPP; Sun et al., 2010) and Advanced automatic QC (AQC) Argo Data version 1.2a (http://www.jamstec.go.jp/ARGO/argo_web/argo/?page_id=100&lang=en). We exclude satellite SSS within 100 km of the coasts, SSH for bottom topography shallower than 200 m, in-situ temperature and salinity duplicated between the GTSPP and AQC Argo, and observations without the best quality flags or whose differences from the forecasts are larger than the values in the gross error check in Table 3. Following Miyazawa et al. (2012) and Penny et al. (2013), the localization scales based on a Gaussian function are chosen to be 300 km and 100 m in the horizontal and vertical directions, respectively. An observational error covariance matrix is assumed to be diagonal using the observation errors in Table 3.

### 3.3. Sensitivity experiments

We conduct sensitivity experiments with combining the IAU and covariance inflation methods [No inflation (NO INFL), RTPP, RTPS, and MULT] to investigate their impacts on the geostrophic balance and accuracy. We set the relaxation parameters in the RTPP and RTPS experiments to $\alpha_{RTPP} = \alpha_{RTPS} = 0.9$ without the IAU and $\alpha_{RTPP} = \alpha_{RTPS} = 0.5, 0.6, ..., 1.2$ with the IAU, and the inflation parameter to $\rho = 1.05^2$ (inflating the forecast ensemble spread by 5%) in the MULT experiments, regardless of the application of the IAU. In this study, we do not explore all values of the relaxation and inflation parameters because of the limitations of computational resources. Hereafter, we refer to the RTPP experiments implemented with and without the IAU as the RTPP+IAU and RTPP experiments, respectively, and the RTPP+IAU experiment with a relaxation parameter of 0.5 as the RTPP05+IAU experiment. Kotsuki et al. (2017) indicated that the RTPP and RTPS do not consider the model error explicitly and that the optimal relaxation parameter may be larger than 1.0. Therefore, we perform experiments with the relaxation parameter >1. To clarify the effects of the IAU and covariance inflation methods, the NO INFL experiment is defined as a control experiment in this study [See Eqs. (9) and (10)].

We integrate the LETKF-based ocean data assimilation system from 7 July 2015 at the start date of the Himawari-8 observations to 31 December 2016, applying the SSS nudging with 90-day timescale to damp a surface freshening drift as in the model spin-up described in subsection 3.1. Furthermore, we conduct 11-day ensemble forecast experiments initialized on

the 1st day of each month in 2016 by the forecasts from the NO INFL, NOINFL+IAU, RTPP09, RTPP09+IAU, RTPS09, RTPS09+IAU experiments, with the SSS nudging applied with 90-day timescale. We estimate $\Delta NBE$ from the ensemble analysis increments on days 1 and 16 of each month, the RMSDs from the daily averaged ensemble mean analyses and forecasts, and the ensemble spread from the daily-mean ensemble analyses. As described in subsection 2.4.2, the statistical analyses are applied to IRs of area-averaged $\Delta NBE$ and analysis RMSDs in all analysis experiments. The results in the RTPP11+IAU and RTPP12+IAU experiments are not shown because numerical instability developed.

## 4. Results

### 4.1 Geostrophic balance

We first compare the geostrophic balance for the various sensitivity experiments using spatiotemporally averaged $\Delta NBE$ over the whole system domain for 2016 (Fig. 1). The NO INFL+IAU experiment has the best geostrophic balance with significant improvement relative to the NO INFL experiment, and thus the IAU plays a role in enhancing the balance, probably because the IAU reduces noise of the high-frequency gravity waves associated with initial shocks. This result is consistent with Yan et al. (2014) who demonstrated that the IAU reduces spurious oscillation of vertical velocity in twin experiments using a relatively idealized EnKF-based ocean data assimilation system. In contrast, since the RTPP09 and RTPS09 experiments show significantly larger $\Delta NBE$ than the NO INFL experiment, the RTPP and RTPS contribute to breaking the balance. The MULT+IAU and MULT experiments give such large $\Delta NBE$ of $2.11 \times 10^{-10}$ and $5.22 \times 10^{-10}$ s$^{-2}$, respectively, that the MULT breaks the balance considerably even if the IAU is applied.

The RTPP+IAU and RTPS+IAU experiments provide significant improvement when the relaxation parameters are $\alpha_{RTPP} \leq 0.9$ and $\alpha_{RTPS} \leq 0.8$, respectively. The RTPP11+IAU experiment becomes numerically unstable in December 2015 and the RTPS11+IAU experiment also significantly degrades the balance, and thus relaxation parameters larger than 1.0 do not appear to be appropriate for the EnKF-based ocean data assimilation system. The combinations of the IAU and RTPP/RTPS, in which the relaxation parameters are set to $\alpha_{RTPP} \leq 0.9$ and $\alpha_{RTPS} \leq 0.8$, appear to maintain geostrophic balance, likely because the IAU counteracts the RTPP/RTPS by improving the balance.

To investigate spatial characteristics of the geostrophic balance, $\Delta NBE$ is temporally averaged over the whole year 2016 (Fig. 2). Here, the RTPP09+IAU and RTPS11+IAU experiments are shown from the RTPP+IAU and RTPS+IAU experiments, because they have the best accuracy as seen in Sect. 4.2. The NO INFL, RTPP09, and RTPS09 experiments produce less balanced fields in the mid-latitude region, especially around the KE (Fig. 2a, c, e). In the RTPS11+IAU experiment, the balance is also lost in higher latitude regions (Fig. 2f). In the MULT and MULT+IAU experiments, there are almost no balanced regions with $\Delta NBE$ smaller than $1.5 \times 10^{-10}$ s$^{-2}$ (figure not shown). The NO INFL+IAU and

RTPP09+IAU experiments show substantial improvement around the KE region, although the relatively large $\Delta NBE$ remains along the KE in the RTPP09+IAU experiment (Fig. 2b, d). Thus, in general, although the balance in the analysis field is not maintained around the KE region, this imbalance is substantially reduced in the NO INFL+IAU and RTPP09+IAU experiments.

## 4.2 Accuracy

### 4.2.1 Surface flow field

We evaluate the accuracy of the surface flow field in the sensitivity experiments, calculating the analysis RMSDs relative to the AVISO observational SSH and SSHA gridded datasets as well as surface zonal and meridional velocity from the drifter buoys. We also estimate the ensemble spread in observational space. As described in Sect. 2.4.3, the AVISO dataset is not independent because it uses satellite SSHAs assimilated in our system, whereas the drifter buoys are independent. The different results from the SSH and SSHA RMSDs are caused by the different MDOT between the AVISO dataset and the system as described in Sect. 2.4.3 and 3.2, respectively. The analysis RMSDs and ensemble spreads are averaged over the whole domain for 2016 in the SSH and SSHA fields (Fig. 3) and the surface zonal and meridional velocity fields (Fig. 4). Compared with the NO INFL experiment, the NO INFL+IAU experiment has significantly larger RMSDs and smaller ensemble spreads in most of the variables, whereas the RTPP09 and RTPS09 experiments show significantly smaller RMSDs and larger spreads (Figs. 3, 4). This indicates that the IAU has a significant effect on reducing the accuracy in the surface flow field because the relatively small ensemble spread leads to small analysis increments and because the IAU does not use the SSH analysis increments. However, the small analysis increments result in a better dynamical balance as shown in subsection 4.1. The result is consistent with Yan et al. (2014) who demonstrated that the IAU degrades accuracy of SSH, temperature, and horizontal velocities using twin experiments. In contrast, the RTPP and RTPS lead to significant improvement by inflating the ensemble spread. The large analysis increments caused by the large ensemble spread might reduce the dynamical balance. The MULT and MULT+IAU experiments yield poor accuracy and very large ensemble spreads in the flow fields; for example, the averaged SSH RMSDs of 0.22 and 0.24 m and the averaged SSH and SSHA ensemble spreads of 0.41 and 0.74 m, respectively. Thus, the MULT does not have sufficient skill in reproducing the flow field.

In both RTPP+IAU and RTPS+IAU experiments, the ensemble spreads are increased in all of the variables for the larger relaxation parameters (Figs. 3c, 4c, d). It appears that the larger relaxation parameters maintain the large ensemble spread induced by the perturbed boundary conditions. In the RTPP+IAU experiments, the accuracy of SSH and SSHA is the highest for $\alpha_{RTPP} = 0.8$, although there is no significant improvement relative to the NO INFL experiment (Fig. 3a, b). The

accuracy in both zonal and meridional velocity improves with larger relaxation parameter, and significantly improves for $\alpha_{RTPP} = 0.8$–$1.0$ (Fig. 4a, b). Consequently, $\alpha_{RTPP} = 0.8$–$0.9$ in the RTPP+IAU experiment may be appropriate to

325 represent the flow field more accurately.

In the RTPS+IAU experiment, the accuracy of the SSH, SSHA, and horizontal velocity tends to improve as the relaxation parameter increases, and then significant degradation suddenly occurs for $\alpha_{RTPS} = 1.2$ (Figs. 3a, b, 4a, b). For $\alpha_{RTPS} = 1.1$, the RTPS+IAU experiment has the best accuracy for the SSH and SSHA but significant improvement only in the SSHA (Fig. 3a, b). For $\alpha_{RTPS} = 1.0$, the accuracy in both zonal and meridional velocity is significantly higher (Fig. 4a,

b). Therefore, $\alpha_{RTPS} = 1.0 - 1.1$ seems to be the best among the RTPS+IAU experiments. We note that the accuracy of the SSH and SSHA in the RTPP+IAU and RTPS+IAU experiments does not surpass the RTPP09 and RTPS09 experiments, probably because the IAU method does not use the SSH analyses. Furthermore, the comparison between the RTPP+IAU and RTPS+IAU experiments suggests that the combination of the IAU and RTPP has higher skill in reproducing the flow field.

To examine spatial features of the analysis accuracy and ensemble spread, the analysis RMSDs and ensemble

spreads in the SSHA are also averaged over 2016 (Figs. 5 and 6, respectively). In most experiments, large RMSDs and ensemble spreads are distributed around the KE region where there are abundant fronts and eddies. Compared with the NO INFL experiment, the ensemble spreads become smaller in the mid-latitude region in the NO INFL+IAU experiment, and thus the accuracy around the KE region is degraded. The RTPP09, RTPS09, RTPP09+IAU, and RTPS11+IAU experiments show larger ensemble spreads leading to improvement of the accuracy around the KE region. However, the larger ensemble

spread is also seen in the subtropical region in the RTPS11+IAU experiment. This does not seem reasonable because a free ensemble experiment does not demonstrate such spread even if the perturbed atmospheric and lateral boundary conditions are applied (figure not shown).

To investigate the forecast accuracy, we calculate the spatiotemporally averaged forecast RMSDs of the 11-day ensemble forecast experiments for each month in 2016 (i.e., total 12 cases) relative to the AVISO and drifter buoys, and the

345 12 cases are averaged to obtain the forecast RMSDs over 2016 (Fig. 7). As shown in Figs. 3, 4 and 7, the results of the forecast RMSDs generally agree with those of the analysis RMSDs, except for the RTPP09+IAU and RTPS09+IAU experiments showing smaller forecast SSHA RMSDs than the NO INFL experiment. Overall, the combination of the IAU and RTPP09 seems to be the most suitable for not only constructing analysis products but also conducting ensemble forecasts.

**4.2.2 The KEO buoy**

We also calculate the analysis and forecast RMSDs relative to independent observations of temperature, salinity, and horizontal velocity from the KEO buoy located south of the KE (Fig. 5a). Here, only the temperature and salinity results are shown since there is basically no improvement in the horizontal velocity. There is almost no difference between the NO INFL+IAU and NO INFL experiments in the temperature analysis accuracy, whereas the salinity analysis accuracy is significantly degraded around 0–200 m depth in the NO INFL+IAU experiment (Fig. 8). Therefore, the IAU may reduce the analysis accuracy, although this is not as obvious as for the flow field shown in Sect. 4.2.1. The RTPP and RTPS experiments give significantly better analysis accuracy than the NO INFL experiment in both temperature and salinity, and thus the RTPP and RTPS play a role in enhancing the analysis accuracy. These results are qualitatively the same as the forecast accuracy (Fig. 9).

When the relaxation parameter $\alpha_{RTPP} = 0.7$–$1.0$ in the RTPP+IAU experiment, the temperature analysis accuracy is significantly enhanced around 200–500 m depth, although there is a slight degradation around 50–150 m depth (Fig. 10a). For the parameter values in that range, the salinity analysis accuracy is also significantly improved in almost all depths (Fig. 10c). Since the temperature analysis accuracy is the best at $\alpha_{RTPP} = 0.8 - 0.9$ and salinity analysis accuracy improves as the relaxation parameter increases, the appropriate relaxation parameter would be $\alpha_{RTPP} = 0.8 - 0.9$ in the RTPP+IAU experiment.

In the RTPS+IAU experiments, the temperature analysis accuracy below 200 m depth is significantly improved for $\alpha_{RTPS} = 0.8 - 1.1$, whereas that above 200 m depth is significantly degraded for $\alpha_{RTPS} = 0.5 - 0.8$ and $\alpha_{RTPS} = 1.2$ (Fig. 10b). The salinity analysis accuracy improves over almost the whole depth when the relaxation parameter is $\alpha_{RTPS} = 1.0$–$1.2$, whereas there is significant degradation around 0–200 m depth when the relaxation parameter is $\alpha_{RTPS} = 0.5$–$0.9$ (Fig. 10d). Therefore, the suitable relaxation parameter is $\alpha_{RTPS} = 1.0 - 1.1$ in the RTPS+IAU experiment.

The RTPP09+IAU and RTPS11+IAU experiments have higher analysis accuracy (Fig. 8), and the RTPP09 and RTPP09+IAU experiments show higher forecast accuracy (Fig. 9) than the other experiments. Therefore, the combination of the IAU and RTPP09 is the most appropriate for the analysis and ensemble forecasts.

## 5. Comparison of the prescribed MULT parameter with the RTPP09+IAU experiment

To investigate how much the inflation in the RTPP09+IAU experiment corresponds to the MULT parameter, we estimate the MULT parameter $\rho_{est}$ corresponding to the RTPP09+IAU experiment using the following equation:

$$\alpha_{RTPP}X^f + (1 - \alpha_{RTPP})X^a_{orig} = \sqrt{\rho_{est}}X^f. \tag{11}$$

By multiplying $(X^f)^T[X^f(X^f)^T]^{-1}$ from the RHS of Eq. (11),

$$\rho_{est}I = \left\{\alpha_{RTPP}I + (1-\alpha_{RTPP})X^a_{orig}(X^f)^T[X^f(X^f)^T]^{-1}\right\}^2, \tag{12}$$

where $I$ denotes the identity matrix. In scalar format, the estimated parameter $\rho_{est}^{(i)}$ at $i$th variable might be represented as

$$\rho_{est}^{(i)} = \left\{\alpha_{RTPP} + \frac{1-\alpha_{RTPP}}{n-1}\frac{X^{a(i)}_{orig}(X^{f(i)})^T}{(\sigma^{f(i)})^2}\right\}^2. \tag{13}$$

Using the outputs from the RTPP09+IAU experiment, we calculate the estimated MULT parameter for the SST, SSS, and SSH fields (Fig. 11). The estimated MULT parameter is large around the mid-latitude region, especially around the Kuroshio Extension region. The estimated MULT parameters averaged over the whole domain and analysis period are 1.08 (1.11) for the SST and SSS (SSH) fields, and these values correspond well to the prescribed MULT parameter $\rho = 1.05^2 \approx 1.10$.

As shown in Fig. 11, the MULT parameter might have the spatial dependency, and therefore adaptive MULT
(Miyoshi, 2011) may be useful. However, Ohishi et al. (in review) demonstrated that the adaptive observation error inflation (AOEI; Minamide and Zhang, 2017; Zhang et al., 2016), with opposite effects to the adaptive MULT, significantly improves dynamical balance and the accuracy of the temperature, salinity, and surface horizontal velocities. This is because the AOEI suppresses the erroneous temperature and salinity analysis increments associated with the representation errors around the Kuroshio Extension region, which result in strong vertical salinity diffusion through weakening density stratification and
therefore degrade the low-salinity structure in the intermediate layer. This implies that the adaptive MULT would increase the analysis increments and degrade the dynamical balance and accuracy. Therefore, it is difficult to find an appropriate MULT parameter.

## 6. Summary
In this study, we have developed an EnKF-based ocean data assimilation system with an assimilation interval of 1 day to take advantage of frequent satellite observations, and we have conducted sensitivity experiments to explore the best combination of the IAU and covariance inflation methods by evaluating the geostrophic balance and analysis accuracy. Table 4 summarizes the overall evaluation in this study. The IAU and RTPP/RTPS have opposite effects to each other; namely, the IAU improves the balance but degrades the accuracy, reducing the ensemble spread, whereas the RTPP and

RTPS degrade the balance and improve the accuracy by inflating the ensemble spread. Large RTPP and RTPS parameters maintain large ensemble spread inflated by the perturbed boundary conditions, and the resulting large analysis increments degrade the balance but improve the accuracy. The RTPP+IAU experiment provides significantly better balance for the relaxation parameters of $\alpha_{RTPP} \leq 0.9$ as well as better accuracy when the relaxation parameter is $\alpha_{RTPP} = 0.8\text{--}1.0$. Therefore, this study demonstrates that the appropriate parameter is $\alpha_{RTPP} = 0.8\text{--}0.9$ when the IAU and RTPP are

combined. In contrast, the RTPS+IAU experiment does not significantly improve the balance and accuracy at the same time, because the balance is significantly better for a relaxation parameter of $\alpha_{RTPS} \leq 0.8$, and the accuracy is significantly higher when the relaxation parameter is $\alpha_{RTPS} = 1.0\text{--}1.1$. Therefore, this study demonstrates that the combination of the IAU and RTPP with the relaxation parameter of $\alpha_{RTPP} = 0.8\text{--}0.9$ is the most suitable for the EnKF-based ocean data assimilation system. The 11-day ensemble forecast experiments show consistent results of forecast accuracy with the analysis accuracy.

In the combination of the IAU and RTPP, the large relaxation parameter of $\alpha_{RTPP} = 0.8\text{--}0.9$ maintains the ensemble spread induced by perturbed boundary conditions and leads to the improvement of the analysis accuracy but degradation of the dynamical balance, and at the same time, the IAU improves the degradation of the dynamical balance by the RTPP. As a result, this would lead to further improvement of the forecast and analysis accuracy by reducing the initial shocks in frequent data assimilation. Compared with the RTPS (RTPS+IAU) experiments, the RTPP (RTPP+IAU)

experiments show better balance and result in smaller initial shocks. As a result, the combination of the IAU and RTPP lead to better accuracy than that of the IAU and RTPS.

The MULT with a 5% inflation of the forecast ensemble spread does not have sufficient skill in maintaining the balance and accurately reproducing the flow field, regardless of whether or not the IAU is applied. Although it is difficult to find an appropriate MULT parameter as described in Section 5, it might be possible that MULT produces analyses with good

balance and accuracy by tuning the inflation parameter. However, since the computational cost of tuning the parameters in all covariance inflation methods is high, this study focuses on the combination of the RTPP/RTPS and IAU with good balance and accuracy. This system still contains other tuning parameters in the perturbed atmospheric forcing, ensemble size, localization scale, and observation errors. We note that the suitable RTPP parameter in the RTPP+IAU experiment would be different depending on those parameter settings. Further experiments are required to determine the best settings for a given

computational resource, and we will address this issue in future studies.

The results of this study would also be useful for constructing EnKF-based data assimilation systems in other fields in which gravity waves have substantial impacts. Furthermore, this study may help improve the accuracy of existing EnKF-based data assimilation systems. Table 1 shows that there are no eddy-resolving EnKF-based ocean reanalysis datasets in the Pacific region. We are now planning to construct such analysis datasets and real-time ensemble prediction systems.

**Code and data availability**

The source codes of the sbPOM and LETKF are available from https://zenodo.org/record/6482744 (last access: 10 May 2022, Jordi and Wang, 2012, doi: 10.5281/zenodo.6482744) and https://github.com/takemasa-miyoshi/letkf (last access: 13 April 2021, Miyoshi and Yamane, 2007), respectively. The source code of the COARE version 3.5 is downloaded 445 from https://github.com/brodeau/aerobulk (last access: 13 April 2021, Brodeau et al., 2017; Edson et al., 2013).

We thank Dr. Kenshi Hibino for providing us with the earlier version of the TE-Global before the official release of the latest version (https://www.eorc.jaxa.jp/water/, last access: 13 April 2021). Details of the observational datasets are as follows: the surface drifter buoys from https://www.aoml.noaa.gov/phod/gdp/hourly_data.php (last access: 13 April 2021, Elipot et al., 2016); the KEO buoy from https://www.pmel.noaa.gov/ocs/ (last access: 13 April 2021); the ETOPO1 from 450 https://www.ngdc.noaa.gov/mgg/global/ (last access: 13 April 2021, Amante and Eakins, 2009); the WOA18 https://www.ncei.noaa.gov/access/world-ocean-atlas-2018/ (last access: 13 April 2021; Locarnini et al., 2019; Zweng et al., 2019); the Himawari-8 satellite SSTs from https://www.eorc.jaxa.jp/ptree/index.html (last access: 13 April 2021; Bessho et al., 2016; Kurihara et al., 2016); the GCOM-W SSTs from https://gportal.jaxa.jp/gpr/?lang=en (last access: 13 April 2021); the satellite SSS from SMOS at http://www.esa.int/Applications/Observing_the_Earth/SMOS (last access: 13 April 2021) 455 and SMAP version 4.3 from https://podaac.jpl.nasa.gov/ (last access: 13 April 2021, Meissner et al., 2018); the satellite SSHA and AVISO (Ducet et al., 2000) from the CMEMS (https://marine.copernicus.eu/, last access: 13 April 2021); and in-situ temperature and salinity from the GTSPP (https://www.ncei.noaa.gov/products/global-temperature-and-salinity-profile-programme, last access: 13 April 2021, Sun et al., 2010) and AQC Argo version 1.2a (https://www.jamstec.go.jp/argo_research/dataset/aqc/index_dataset.html, last access: 13 April 2021). The global JRA55 460 atmosphere and SODA 3.7.2 ocean reanalysis datasets are from http://search.diasjp.net/en/dataset/JRA55 (last access: 13 April 2021, Kobayashi et al., 2015) and https://www.soda.umd.edu/soda3_readme.htm (last access: 13 April 2021, Carton et al., 2018), respectively.

**Author contributions**

SO, TH, and YM developed the code of the ocean data assimilation system. SO conducted the sensitivity experiments and analyzed their outputs. SO and TM prepared the paper with contributions from all coauthors (TH, HA, JI, YM, and MK).

**Competing interests**

The authors declare that they have no conflict of interest.

**Acknowledgments**

We thank Dr. Yue Ying and two anonymous reviewers for their constructive comments. We are very grateful to Dr. Shunji Kotsuki at Chiba University for providing us with sample code of the RTPP and RTPS. Numerous comments from Drs. Nariaki Hirose, Takahiro Toyoda, Yosuke Fujii, and Norihisa Usui at the Meteorological Research Institute, Yoichi Ishikawa at JAMSTEC, Katsumi Takayama at IDEA Consultants, Inc., Naoki Hirose at Kyushu University, and participants in the ocean data assimilation summer school also helped us develop the system. This work used computational resources of the JAXA Supercomputer System Generation 2 and 3 (JSS2 and JSS3, respectively) and the supercomputer Fugaku provided by RIKEN through the HPCI Research Project (Project ID: hp210166, hp220167, ra000007).

**Financial support**

This work was supported by JST AIP Grant Number JPMJCR19U2, Japan; MEXT (JPMXP1020200305) as "Program for Promoting Researches on the Supercomputer Fugaku" (Large Ensemble Atmospheric and Environmental Prediction for Disaster Prevention and Mitigation); the COE research grant in computational science from Hyogo Prefecture and Kobe City through Foundation for Computational Science; JST, SICORP Grant Number JPMJSC1804, Japan; JSPS KAKENHI Grant Number JP19H05605; the Japan Aerospace Exploration Agency (JX-PSPC-452680, -500973, -509736, -513414, -519799, and -527843); JST, CREST Grant Number JPMJCR20F2, Japan; Cabinet Office, Government of Japan, Moonshot R&D Program for Agriculture, Forestry and Fisheries (funding agency: Bio-oriented Technology Research Advancement Institution) No. JPJ009237; RIKEN Pioneering Project "Prediction for Science"; JST CREST Grant Number JPMJSA2109.

**Review statement:** This paper was edited by Dr. Yuefei Zeng and reviewed by Dr. Yue Ying and two anonymous reviewers.

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

**Table 1: Overview of EnKF-based ocean data assimilation systems developed after 2010. Abbreviations are PEODAS: Predictive Ocean Atmosphere Model for Australia (PAOMA) Ensemble Ocean Data Assimilation system; DEnKF: Deterministic EnKF (Sakov and Oke, 2008); LETKF: Local ensemble transform Kalman filter (Hunt et al., 2007); EAKF: Ensemble adjustment Kalman filter (Anderson, 2001); LESKTF: Local Error Subspace Kalman Transform Filter (Nerger et al., 2012); T: Temperature; S: Salinity; SST: Sea surface temperature; SSH: Sea surface height; and MULT: Multiplicative inflation. Adaptive MULT was proposed by Miyoshi (2011). Dashes are used to indicate no application. "Inflated obs. error" in TOPAZ4 indicates that observation errors are inflated when ensemble analyses are calculated.**

| Name | PEODAS (Yin et al., 2011) | TOPAZ4 (Sakov et al., 2012) | Miyazawa et al. (2012) | Karspeck et al. (2013) | Penny et al. (2013) | Penny et al. (2015) | Baduru et al. (2019) | Brüning et al. (2021) |
|---|---|---|---|---|---|---|---|---|
| Domain | Global | North Atlantic +Arctic (Release) | South of Japan | Global | Quasi global | Global | Indian Ocean | North Sea +Baltic Sea |
| Horizontal resolution (longitude × latitude) | 2° × 0.5–1.5° | 12–16km × 12–16km | 1/36° × 1/36° | 1° × 1° | 1° × 0.58–1° | 0.5° × 0.25–0.5° | 1/12° × 1/12° | 0.9–5km× 0.9–5km |
| Vertical resolution | 25 z-levels | 28 hybrid layers | 31 σ-layers | 60 z-levels | 20 z-levels | 40 z-levels | 40 σ-layers | 25–36 layers |
| Perturbed boundary condition | Atmosphere | Atmosphere | – | – | Atmosphere | Atmosphere | Atmosphere | – |
| EnKF | Simplified EnKF | DEnKF | LETKF | EAKF | LETKF | LETKF | LETKF | LESKTF |
| Ensemble size | 11 | 100 | 20 | 48 | 40 | 28 | 80 | 12 |
| Assimilation window | 5 days | 7 days | 2 days | 1 day | 5 days | 5 days | 5 days | 12 hours |
| Assimilated data | T, S | SST, SSH, T, S, Ice | SST, SSH, T, S | T, S | T, S | T, S | SST, T, S | SST |
| Covariance inflation | Additive inflation | – | MULT | – | Adaptive MULT | – | MULT | – |
| IAU/Nudging | – | – | – | – | – | – | – | – |
| Period | 1979–2006 | 1991–2019 | 2010.02.08–28 | 1998–2005 | 1997–2003 | 1991–98 | 2016.08–2018.09 | 2021– |
| Other | | Inflated obs. error | | | | | | |

**Table 2: Overview of the regional ocean model in the ocean data assimilation system**

| | |
|---|---|
| Ocean model | sbPOM (Jordi and Wang, 2012) |
| Model domain | Northwestern Pacific [117°–180°E, 15°–50°N] |
| Horizontal resolution | 0.25°×0.25° |
| Vertical layer | 50 σ-layers |
| Initial conditions | WOA18 (Locarnini et al., 2019; Zweng et al., 2019) |
| Atmospheric forcing | JRA55 (Kobayashi et al., 2015) |
| River discharge | TE-Global (https://www.eorc.jaxa.jp/water/) |
| Lateral boundary condition | SODA version 3.7.2 (Carton et al., 2018) |
| Spin-up period | 2011.01.01–2015.07.06 |

**Table 3: Overview of data assimilation in the ocean data assimilation system**

| | |
|---|---|
| Assimilation method | LETKF (Hunt et al., 2007; Miyoshi and Yamane, 2007) |
| Ensemble size | 100 |
| Assimilation cycle | 1 day |
| Observations | |
|   SST | Himawari-8 (Bessho et al., 2016; Kurihara et al., 2016) |
| | GCOM-W (http://www.ghrsst.org) |
|   SSS | SMOS (https://earth.esa.int) |
| | SMAP (https://www.jpl.nasa.gov) |
|   SSH | |
|     - SSHA | DUACS multimission satellite data (https://marine.copernicus.eu/) |
|     - MDOT | Climatology of model outputs in a spin-up period (2012–14) |
|   Temperature and Salinity | GTSPP (Sun et al., 2010) |
| | AQC Argo data version 1.2a |
| | (http://www.jamstec.go.jp/ARGO/argo_web/argo/?page_id=100&lang=en) |
| Horizontal localization scale | 300 km |
| Vertical localization scale | 100 m |
| Observation error | |
|   SST | 1.5°C |
|   SSS | 0.3 |
|   SSH | 0.2 m |
|   Temperature | 1.5°C |
|   Salinity | 0.3 |
| Gross error check | |
|   SST | $\pm 5$ °C |
|   SSS | $\pm 1$ |
|   SSH | $\pm 1$ m |
|   Temperature | $\pm 5$°C |
|   Salinity | $\pm 2$ |
| Assimilation period | 2015.07.07–2016.12.31 |

**Table 4: Schematic summarizing the evaluation of the geostrophic balance and analysis accuracy of the AVISO SSH and SSHA, the surface zonal and meridional velocity from the drifter buoys, and the temperature and salinity at the KEO buoy in the sensitivity experiments. Open circles and crosses indicate improvement and degradation relative to the NO INFL experiment, respectively, and asterisks denote significant improvement and degradation. In rows two to four, symbols and asterisks are used only if both variables have the same results; otherwise, dashes are used to indicate no significant difference from the NO INFL experiment. Parentheses in the RTPP+IAU and RTPS+IAU experiments denote the best relaxation parameter in the second row and the range of the relaxation parameter with significant improvement in the other rows.**

| | IAU | RTPP09 | RTPS09 | RTPP+IAU | RTPS+IAU |
|---|---|---|---|---|---|
| Geostrophic balance | ○* | ×* | ×* | ○* (Sig. at $\alpha_{RTPP} \leq 0.9$) | ○* (Sig. at $\alpha_{RTPS} \leq 0.8$) |
| SSH and SSHA from the AVISO | ×* | ○* | ○* | – (Best at $\alpha_{RTPP} = 0.8$) | – (Best at $\alpha_{RTPS} = 1.1$) |
| Surface velocity from the drifter buoys | ×* | ○ | ○* | ○* (Sig. at $\alpha_{RTPP} = 0.8 - 1.0$) | ○* (Sig. at $\alpha_{RTPS} = 1.0$) |
| T and S at the KEO buoy | × | ○* | ○* | ○* (Sig. at $\alpha_{RTPP} = 0.7 - 1.0$) | ○* (Sig. at $\alpha_{RTPS} = 1.0 - 1.1$) |

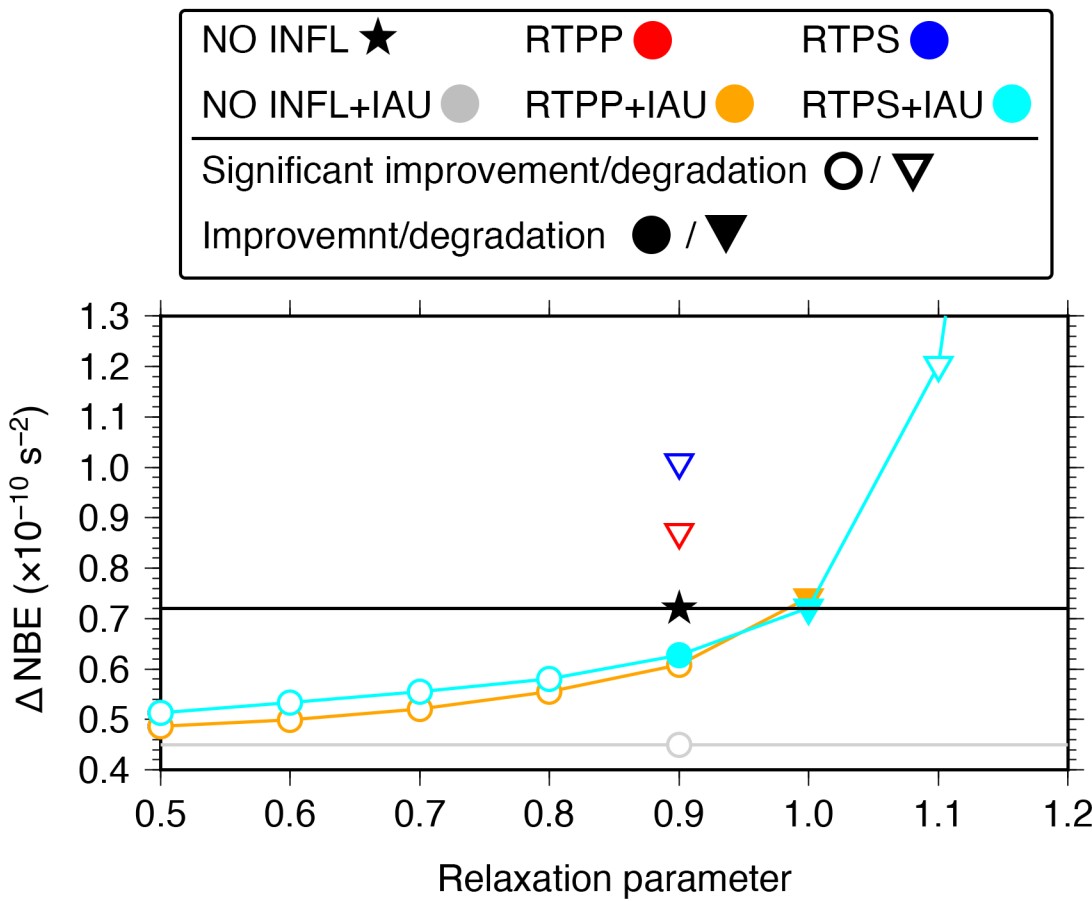

**Figure 1: Spatiotemporally averaged ∆*NBE* over the whole domain for 2016 in the NO INFL (black star), RTPP (red), RTPS (blue), NO INFL+IAU (gray), RTPP+IAU (orange), and RTPS+IAU (cyan) experiments as a function of the relaxation parameters. Open circles and triangles indicate significant improvement and degradation relative to the NO INFL experiment at a 99% confidence level, respectively, and closed circles and triangles denote improvement and degradation of NO INFL experiment with no significant differences. The RTPS12+IAU, MULT+IAU, and MULT experiments show significant degradation with the averaged ∆*NBE* of 2.94, 2.11, and 5.22 × 10⁻¹⁰ s⁻², respectively (not shown). The RTPP+IAU experiments for the relaxation parameters of $\alpha_{RTPP} \geq 1.1$ are not shown because numerical instability developed.**

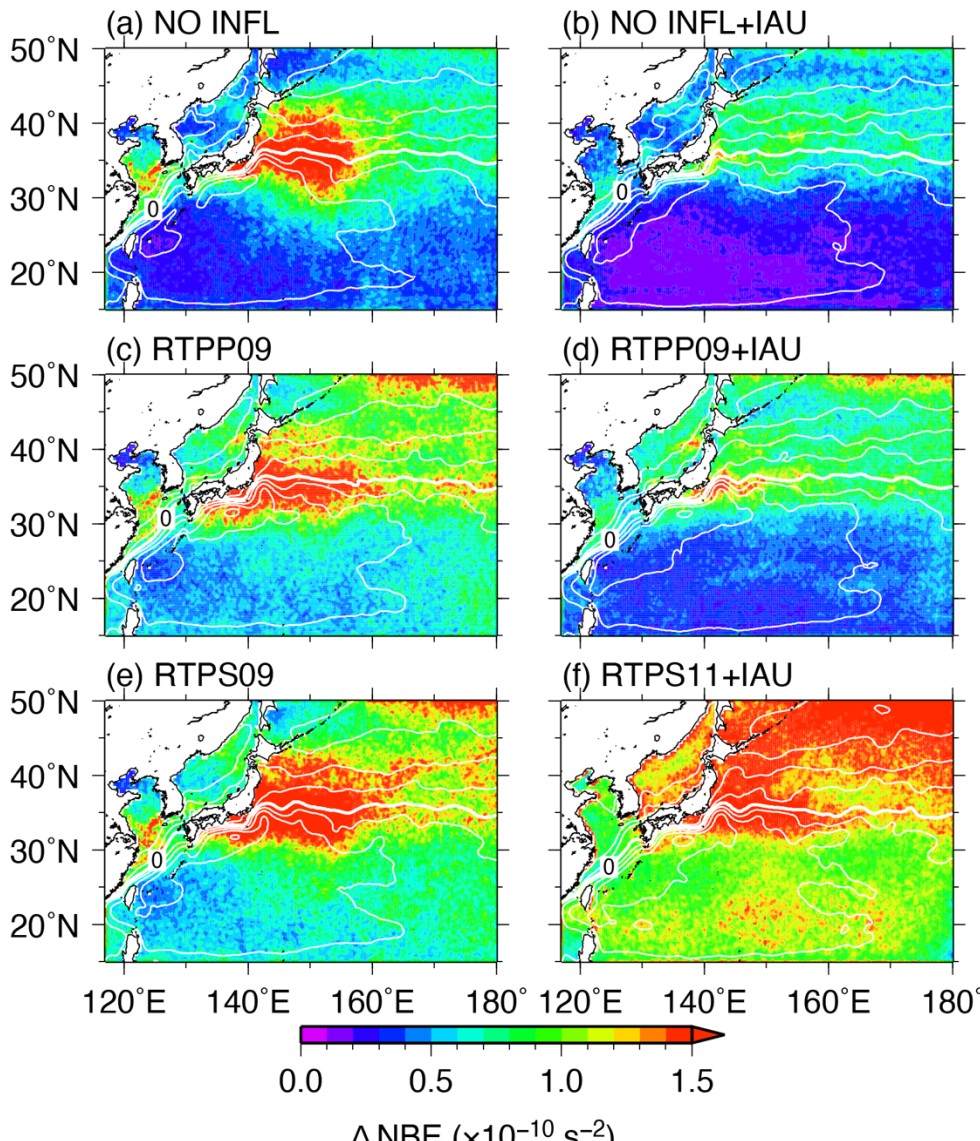

**Figure 2:** Δ*NBE* (colors) and SSH (white contours) averaged over 2016 in the (a) NO INFL, (b) NO INFL+IAU, (c) RTPP09, (d) RTPP09+IAU, (e) RTPS09, (f) RTPS11+IAU experiments. Thin (thick) contour intervals are 0.2 m (1.0 m).

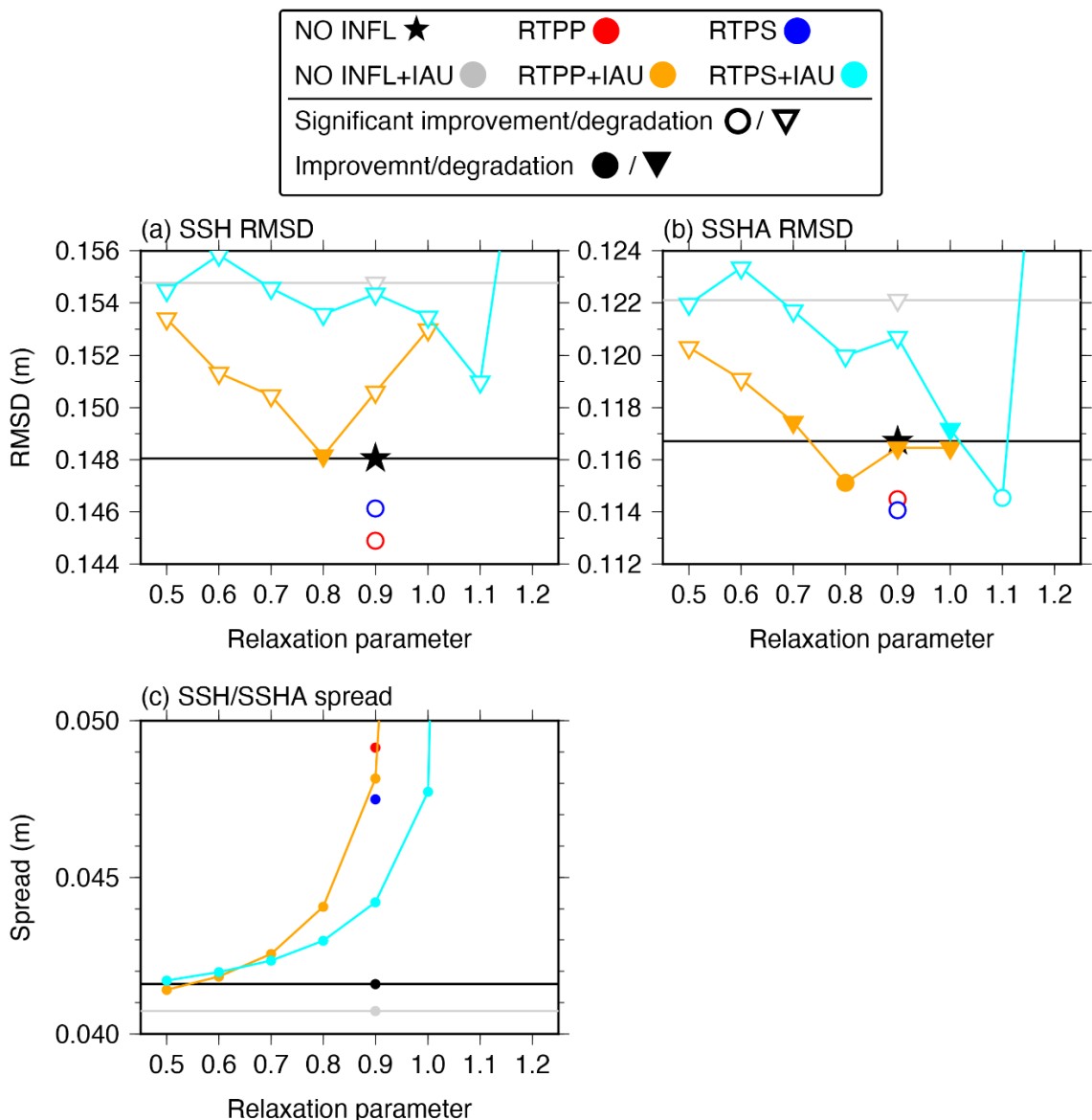

**Figure 3: As in Fig. 1 but for the analysis RMSDs of (a) SSH and (b) SSHA relative to the AVISO dataset. (c) Spatiotemporally averaged ensemble spreads of SSH and SSHA over the whole domain for 2016 in observational space (circles). The RMSDs of SSH and SSHA in the RTPS12+IAU experiment are 0.164 and 0.137 m, respectively (not shown).**

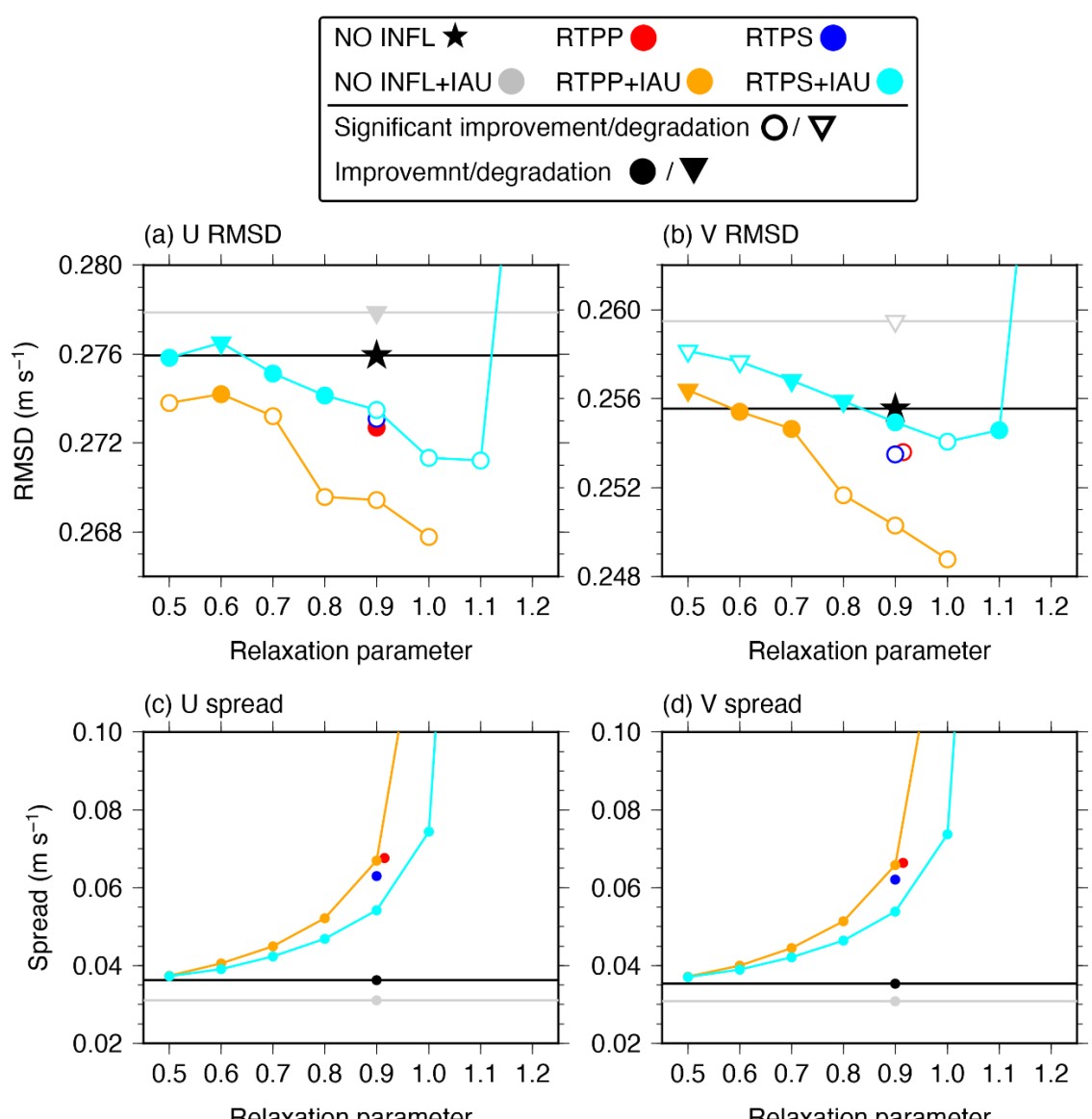

**Figure 4: As in Fig. 1 but for the analysis RMSDs of the surface (a) zonal and (b) meridional velocity relative to the**
**drifter buoys, and the ensemble spreads of the surface (c) zonal and (d) meridional velocity. The RMSDs of surface**
**zonal and meridional velocity in the RTPS12+IAU experiment are 0.293 and 0.277 m s⁻¹, respectively (not shown).**
**The RMSD in (b) and ensemble spreads in (c) and (d) in the RTPP09 experiment are slightly offset for visualization.**

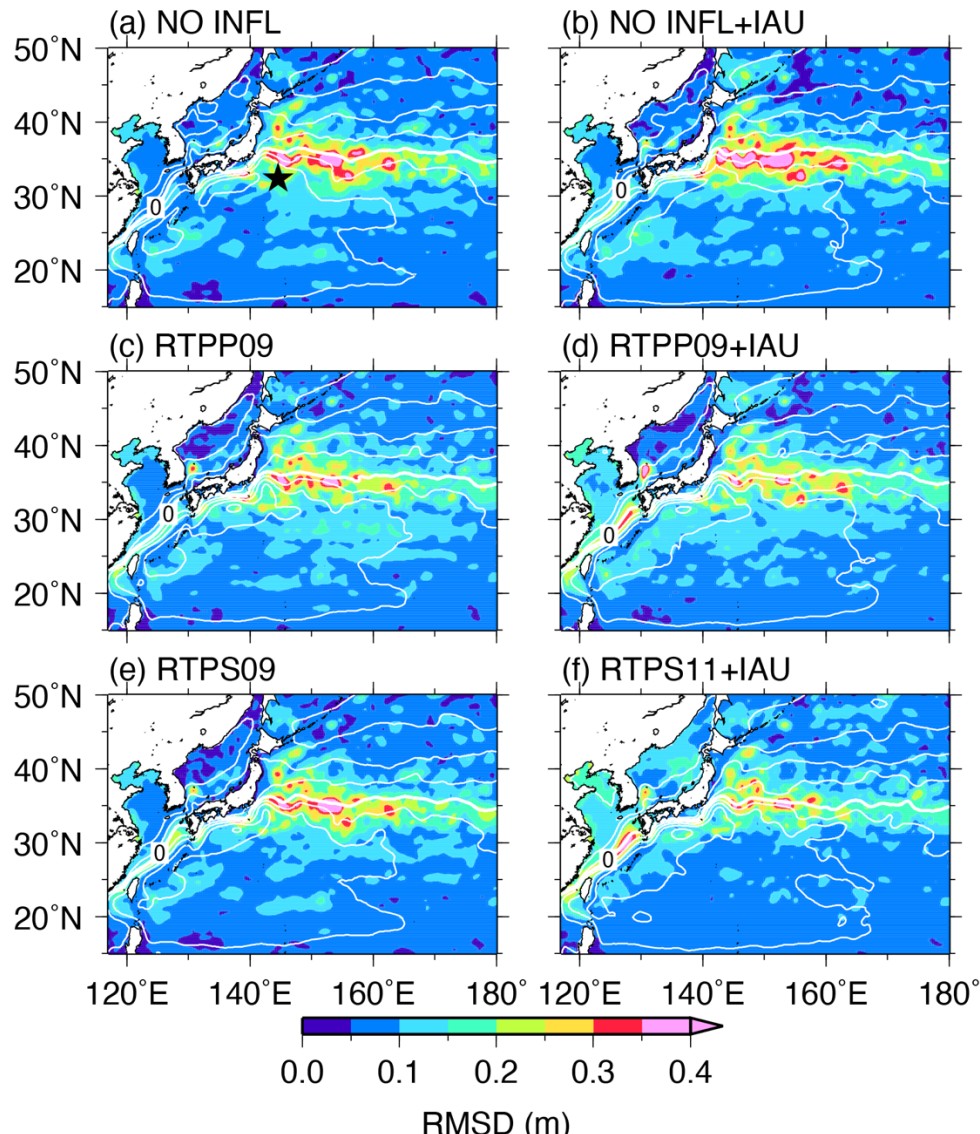

**Figure 5: As in Fig. 3 but for the analysis RMSDs relative to the SSHA from the AVISO dataset (color). Black star in**
**(a) indicates the KEO buoy location (144.6°E, 32.3°N).**

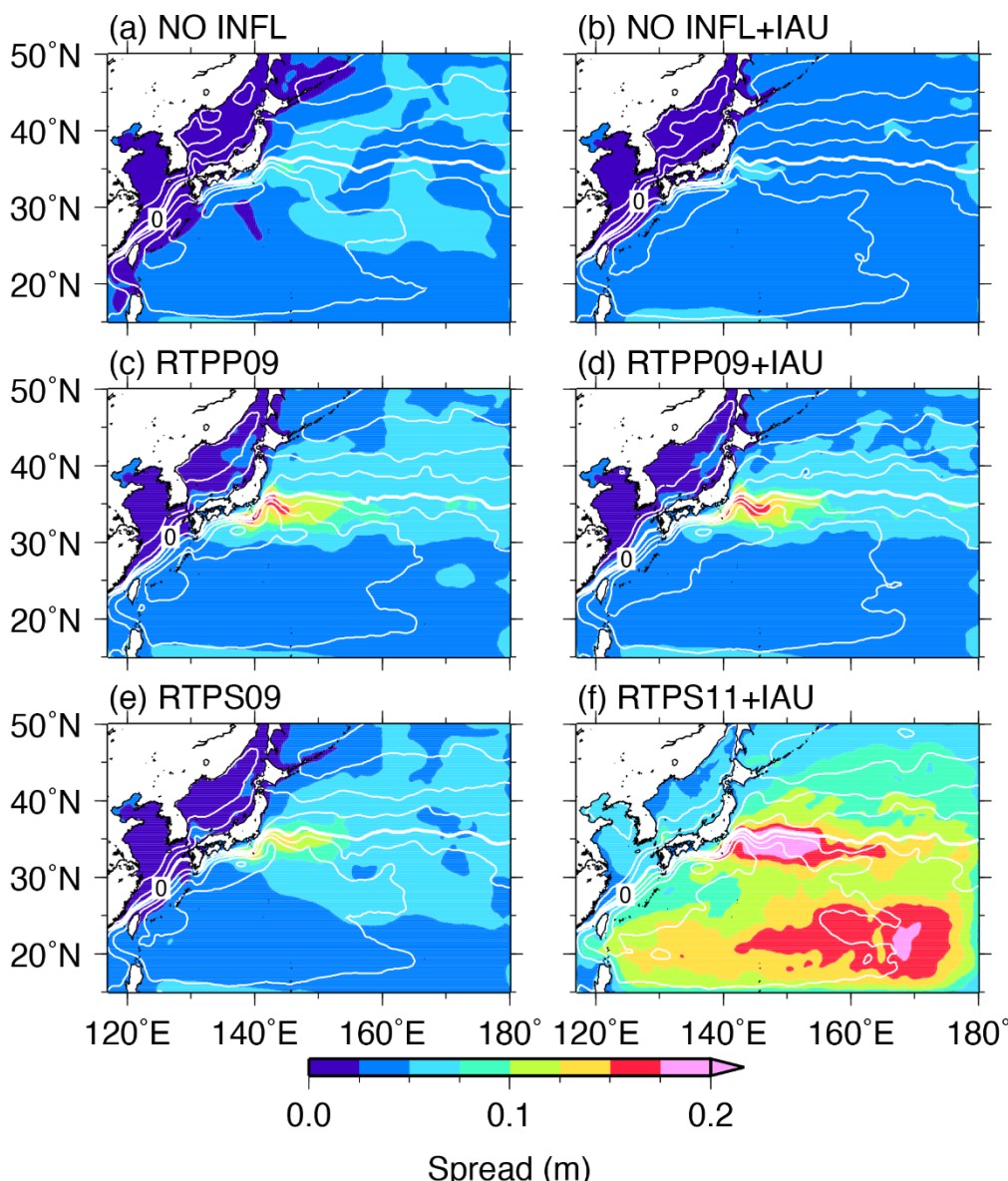

**Figure 6: As in Fig. 3 but for the SSHA ensemble spreads.**

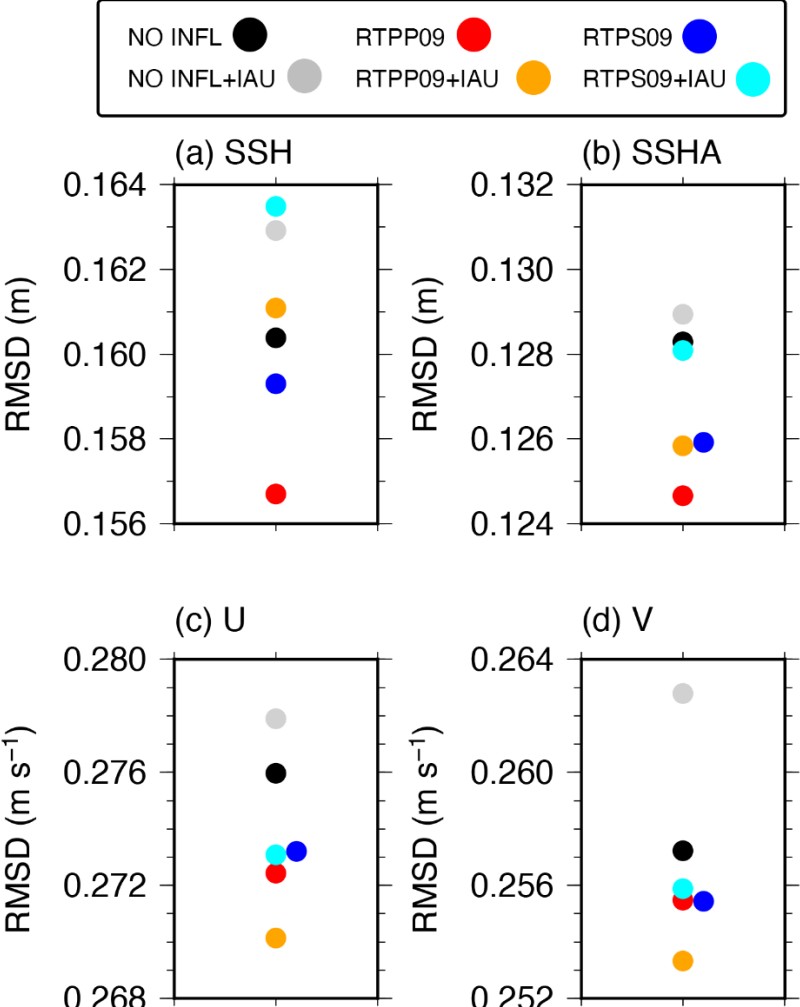

**Fig. 7 Spatiotemporally averaged RMSDs of 11-day ensemble forecast in the NO INFL (black), RTPP09 (red), RTPS09 (blue), NO INFL+IAU (gray), RTPP09+IAU (orange), and RTPS09+IAU (cyan) experiments relative to surface (a) SSH and (b) SSHA from the AVISO and (c) zonal and (d) meridional velocities from the drifter buoys.**

**The RMSDs in the RTPS09 in (b), (c), and (d) are slightly offset for visualization.**

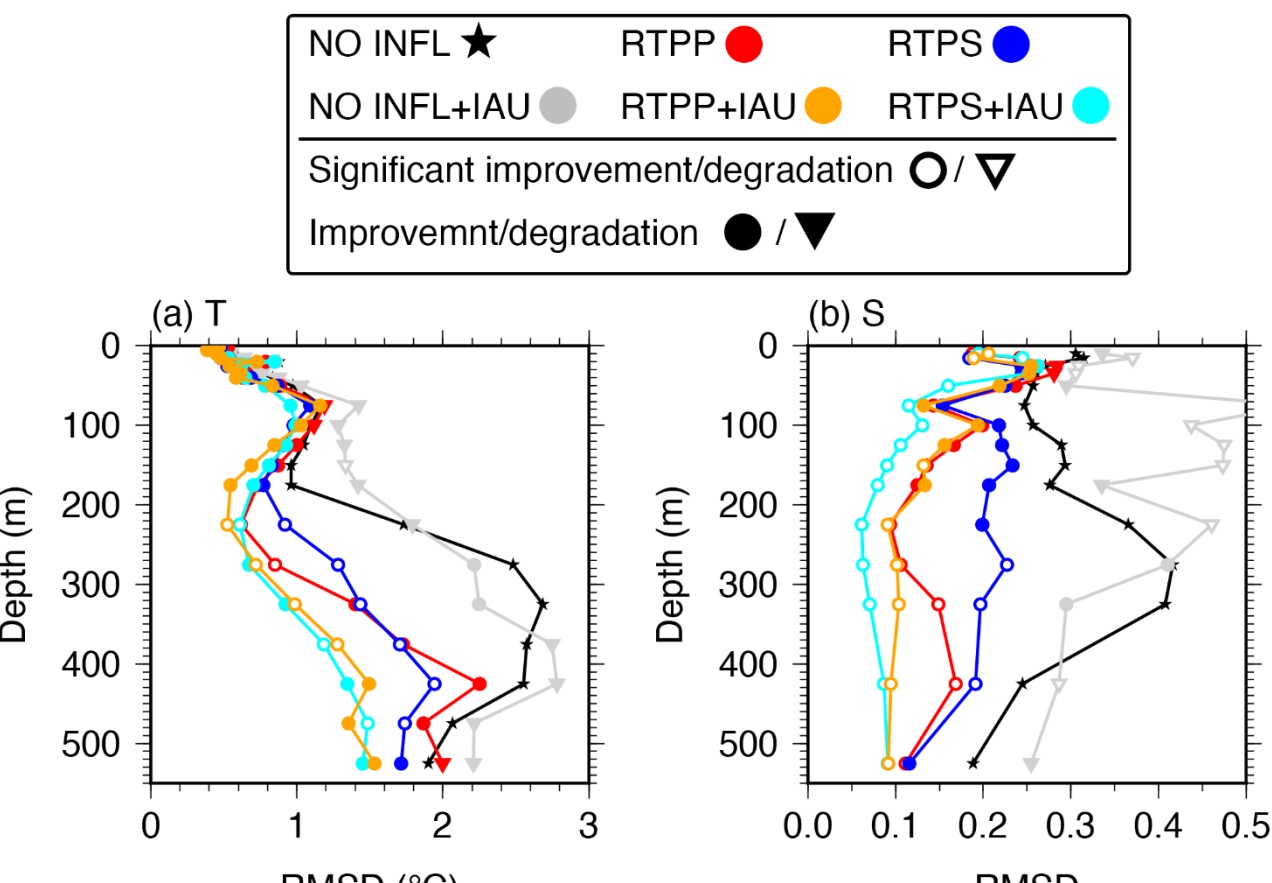

**Figure 8: Analysis RMSDs of (a) temperature and (b) salinity relative to the KEO buoy averaged over 2016 in the NO INFL (black star), RTPP09 (red), RTPS09 (blue), NO INFL+IAU (gray), RTPP09+IAU (orange), RTPS11+IAU (cyan) experiments. Open circles and triangles denote significant improvement and degradation relative to the NO INFL experiment at a 99% confidence level, respectively. Closed circles and triangles indicate improvement and degradation with no significant differences, respectively.**

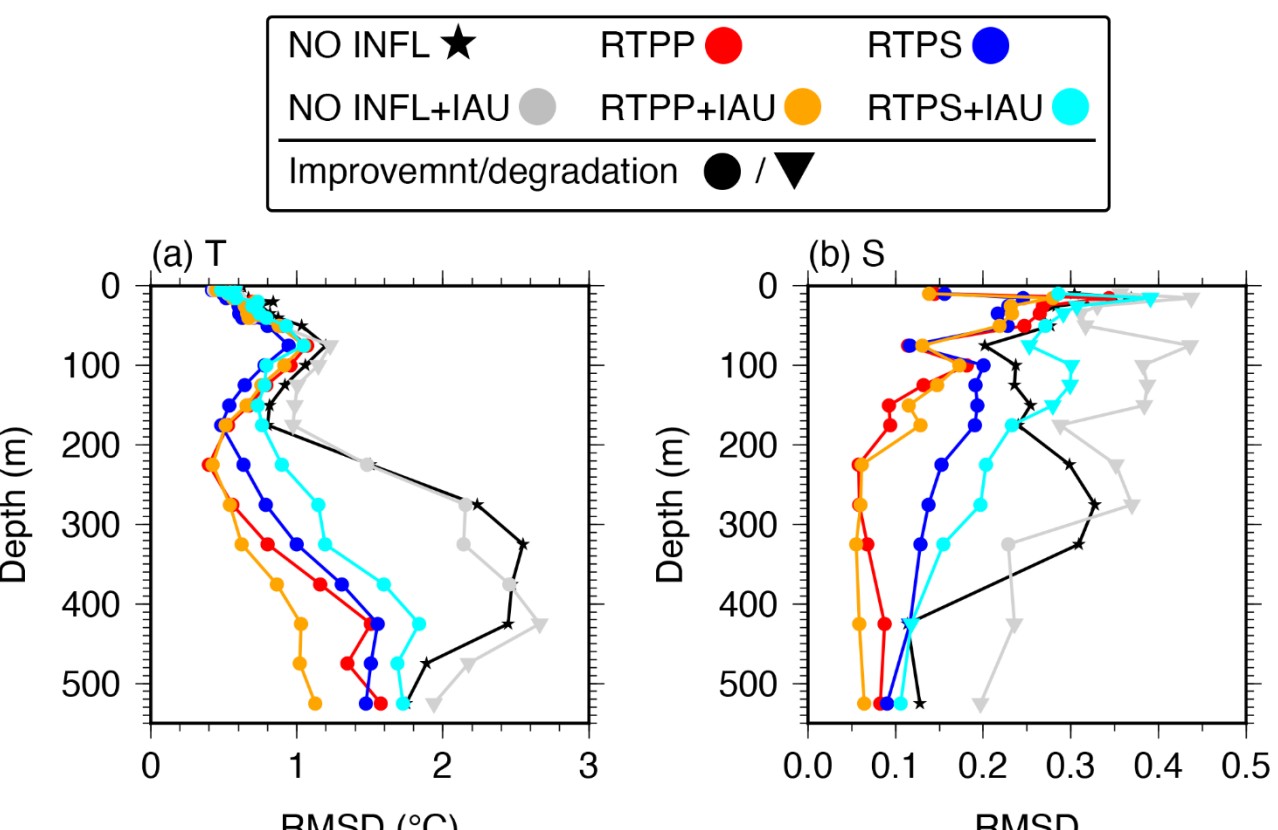

**Figure 9: As in Fig. 8 but for the forecast RMSDs of the 11-day ensemble forecast experiments. Here, we note that the results of the RTPS09+IAU experiment are shown here, whereas the analysis RMSDs of the RTPS11+IAU experiment are shown in Fig. 8, and thus the relaxation parameters are different in the RTPS+IAU experiments.**

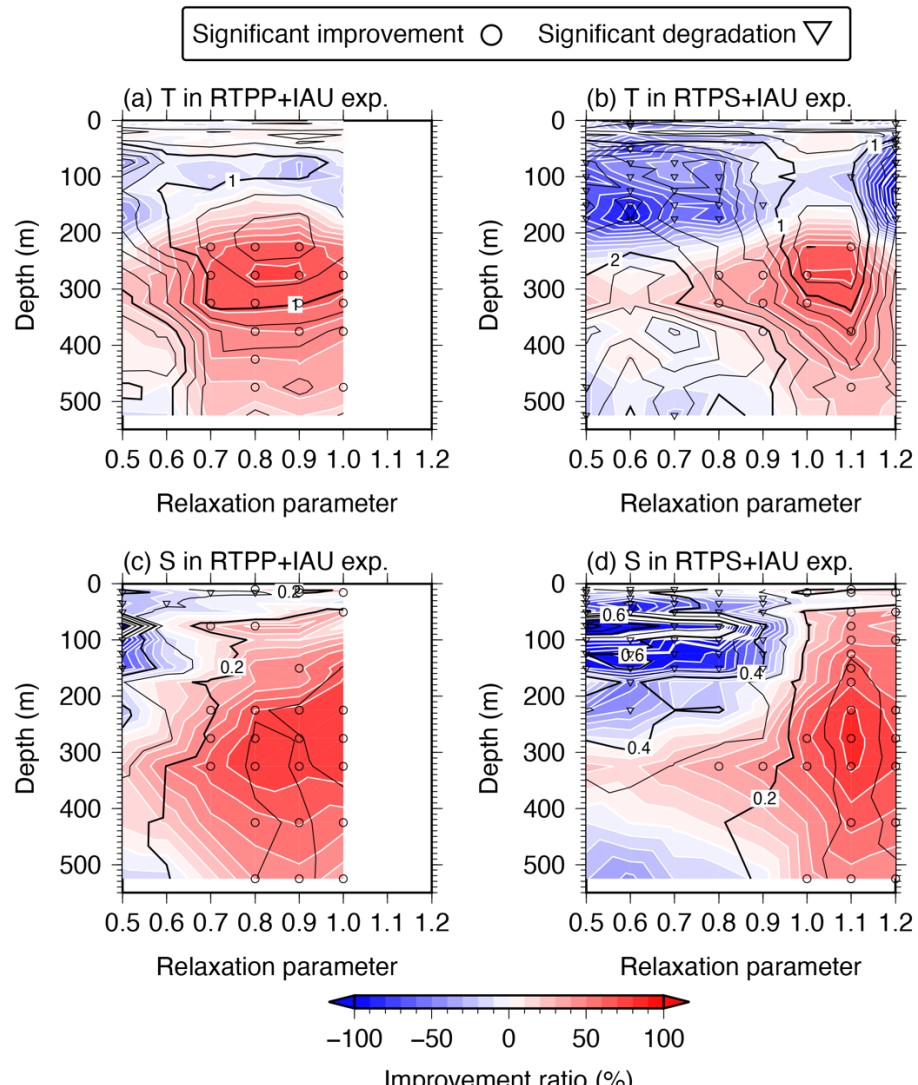

**Figure 10: Temperature analysis RMSDs (black contours) and IRs (color shading and white contours) between the KEO buoy and (a) RTPP+IAU and (b) RTPS+IAU experiments averaged over 2016. (c) and (d) as in (a) and (b) but for salinity. Open circles and triangles indicate significant improvement and degradation relative to the NO INFL experiment at a 99% confidence level, respectively. Thin (thick) black contour intervals are 0.2 (1.0) °C in (a) and (b), and 0.1 (0.2) in (c) and (d); thin (thick) white contour intervals are 10% (100%).**

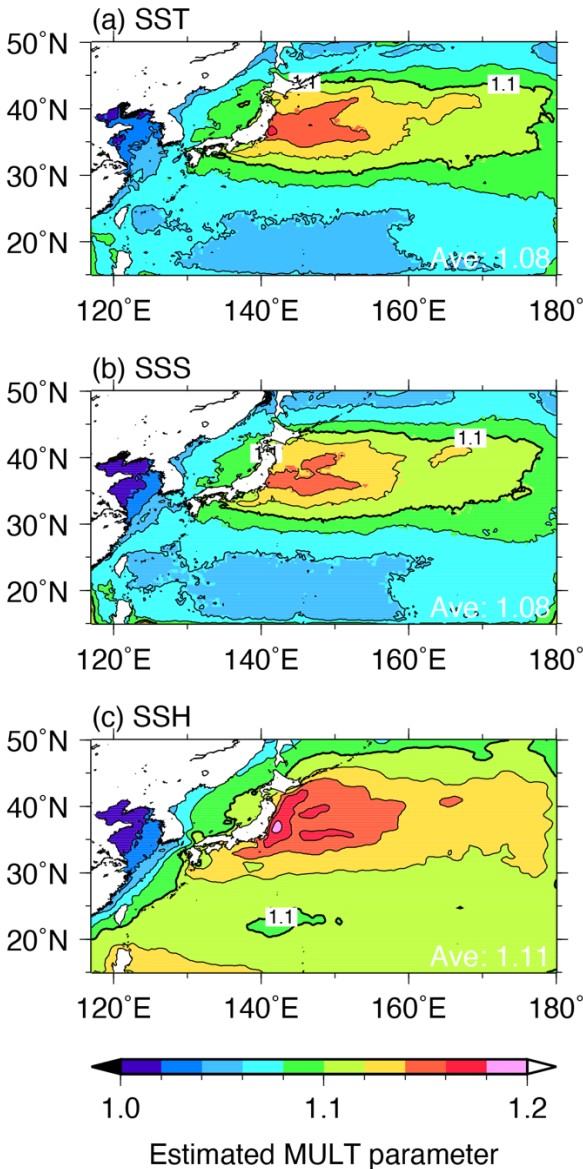

**Figure 11: Estimated MULT parameters [Eq. (13)] averaged over 2016 for (a) SST, (b) SSS, and (c) SSH fields using the outputs from the RTPP09+IAU experiment. Right bottom values indicate spatiotemporally averaged estimated MULT parameters. Thin (Thick) counter intervals are 0.02 (0.1).**