# Peer review of "An ensemble Kalman filter system with the Stony Brook Parallel Ocean Model v1.0"

_Geoscientific Model Development, 2022_

## Author Response (AR1)

■ **Reviewer 1**

Summary: This paper documents the tuning of relaxation parameters for an ocean DA based on EnKF and IAU. The methods used in this study are well established in other DA practice. Since ocean DA faces the challenge from dynamical imbalance with shorter cycling periods, the use of relaxation with IAU is a good approach and tuning results are meaningful for the ocean prediction community. I've found several issues in experimental design and result presentation, which I believe the authors should be able to address before the paper can be accepted.

We thank the reviewer for constructive comments, especially on the MULT parameter. We have investigated how much the inflation in the RTPP09+IAU experiment corresponds to the MULT parameter, as replied to the second major comment.

Major Issues:

1. Are cases with alpha<0.9 tested for RTPP/RTPS without IAU? As RTPP+IAU approaches NO INFL+IAU results in both balance and accuracy, I guess the RTPP cases will also approach NO INFL as alpha decrease. If you have tested several points (maybe RTPP05 and RTPP07) it would be interesting to add them in the plots. For example, could there be an alpha value for RTPP that beats RTPP09+IAU?

As described in subsection 3.3, we have performed the RTPP and RTPS experiments without the IAU only for $\alpha_{RTPP} = \alpha_{RTPS} = 0.9$ because of the limitation of computational resources. Since RTPP+IAU and RTPS+IAU experiments gradually approach the NO INFL+IAU experiment as the relaxation parameters decreases, it is likely that the RTPP and RTPS experiments without IAU also gradually approach the NO INFL experiment. Therefore, the RTPP and RTPS experiments would not surpass both dynamical balance and accuracy in the RTPP09+IAU experiment. We would appreciate your understanding of the matter of computational resources.

2. The choice of inflation factor in MULTI is more problematic. Since the multiplicative inflation is applied throughout the domain, it is more sensitive the the rho value. The relaxation methods have build-in spatial variations in inflation so I think it is not a fair comparison between MULTI and RTPP/RTPS methods. In regions where analysis increments are smaller (fewer observations) the inflation of spread can accumulate over time exponentially. Ideally using a spatial varying inflation (such as in adaptive MULTI algorithms) can help. So, if you choose to show MULTI results here the exact value of

rho is very important. Could you estimate an equivalent rho from the best RTPP/RTPS cases? You can averaged the (1-alpha) + alpha*prior_spread/posterior_spread over the domain and time (for RTPP) to estimate the equivalent rho, is it near 1.05 or much smaller?

We thank the reviewer for constructive comments on MULT. We have added the results of the estimated MULT parameter corresponding to the RTPP09+IAU experiment and the discussion of the adaptive MULT, respectively, to the first and second paragraph in Section 5 in the revised manuscript. In the first paragraph, we have indicated that the estimated MULT parameters for the SST, SSS, and SSH averaged over the whole analysis period and domain ($\rho_{est} = 1.08, 1.08, 1.11$, respectively) correspond well to the prescribed parameter ($\rho = 1.05^2 \approx 1.10$). In the second paragraph, we have described that the adaptive MULT would result in degradation, because Ohishi et al. (in review) demonstrated that adaptive observation error inflation (AOEI; Minamide and Zhang 2017), with opposite effects to the adaptive MULT, significantly improves dynamical balance and accuracy of temperature, salinity, and surface horizontal velocities.

As indicated by the reviewer, observations in the ocean interior are relatively sparse, and the ensemble spread might be exponentially inflated over time in MULT experiments. Consequently, the MULT might not be suitable for ocean data assimilation. However, we could not deny that appropriate MULT parameters do not exist. Therefore, we have described "Although it is difficult to find appropriate an MULT parameter as described in Section 5, it might be possible that MULT produces analyses with good balance and accuracy by tuning the inflation parameter." in the second paragraph in Section 6 in the revised manuscript.

3. Tuning of relaxation can also be case-dependent, you also need to consider the density of observations and localization radius. In the method description maybe you should state more clearly how you tuned localization with this observing network (can you also show a map of observation density for reference?), and the results from tuning alpha in relaxation would likely change if one use another set of observations with different density and localization radius. A discussion in the conclusion would be nice.

Figure R1 shows the frequency of in-situ observations at 5° longitude×5° latitude bins in 2016. Except for coastal regions, 30 observations per month are broadly distributed over the whole domain, and there is no gap in the observation density over the offshore regions.

As discussed in subsection 3.2 in the original and revised manuscripts, we have prescribed the localization scale following Miyazawa et al. (2012) and Penny et al. (2013). The localization scale is not optimal and required for tuning, but this is an issue for future studies as described in the second paragraph in Section 5 (6) in the original (revised) manuscript.

As indicated by the reviewer, it is better to notice that readers are necessary to tune the RTPP parameters for their experimental setting. Consequently, we have added the discussion to the second paragraph in Section 6 in the revised manuscript.

4. I found time evolution of imbalance and errors to be important in this particular case. Since you used fixed values in inflation schemes, it is not guaranteed that the performance will be steady in time. Does the imbalance gradually increase or decrease over time for a chosen alpha value? A time series of spatially averaged delta NBE could be more convincing that the performance is steady. I would be also curious about how long the initial spin up period is for DA solutions to become steady.

The RTPP09+IAU experiment shows that it is not clear whether $\Delta NBE$ averaged over the domain have substantial trends or not (Fig. R2). Rather, it appears to undergo seasonal variations with larger (smaller) $\Delta NBE$ in summer (winter). This is the same for the other experiments except for the MULT+IAU and MULT experiments. We might be able to obtain insights into the spin-up period if we conduct the experiments for a longer period, but this is out of the scope of this study.

Minor Issues:

We thank the reviewer for carefully checking the manuscript. We have modified almost parts following your comments.

Line 153: abs denotes taking the absolute value, please use standard notation |x|.

We have modified the notation for the absolute function in Eq. (8).

Line 160: the same term IR is used for both RMSE and NBE?, maybe add a suffix to distinguish.

We have added suffix N and R to IR for $\Delta NBE$ and RMSD in Eqs. (9) and (10),

respectively.

Line 220, Table 3: gross error check not "growth error check"?

 We have replaced "growth error check" with "gross error check" in subsection 3.2 and Table 3.

Line 237: SSS nudging: could you provide more details of this approach, maybe a reference or technical report?

 The SSS nudging during the data assimilation experiments is the same as the model spin-up. We have added "as in the model spin-up described in subsection 3.1" to the end of the first sentence of the second paragraph in subsection 3.3 in the revised manuscript.

Line 246: is every experiment tested against NO INFL for significance of improvement/degradation? If so, you should state this more clearly.

 We have clarified the detail of the statistical analysis by modifying the descriptions in subsection 2.4.2 and the 2nd paragraph in subsection 3.3.

Line 264: this imbalance is substantially improved => reduced.

 We have replaced "improved" with "reduced" in the last sentence of subsection 4.1.

Line 276: I guess the RTPP09 and RTPS09 cases are also tested against NO INFL for significance?

 We have conducted statistical analyses for the RTPP09 and RTPS09 experiments relative to the NO INFL experiment. We have clarified the description of the statistical method, as is replied to the above comment regarding Line 246.

Figures 1, 3, 4, 7 and 8: you used hollow/solid circles to denote significant/non-significant improvements, but for degradation you used "x" which cannot show hollow/solid differences, maybe use another symbal (triangle?) so you can be consistent.

We have used closed circles for no significant improvement and degradation, open circles for significant improvement, and cross marks for significant degradation throughout the manuscript. If different symbols are used for no significant improvement and degradation, the figures might be hard to see. Therefore, we have maintained the symbols.

Line 551: confidence limit: do you mean confidence level (p value < 0.01)? and no significant difference has p>0.01?, if you used t-test just state the p value threshold to be clear.

We have replaced "confidence limit" with "confidence level" in the caption of Fig. 1. As described in subsection 2.4.2 in the original and revised manuscripts, we have used a bootstrap method rather than a t-test because of difficulties to accurately estimate the degree of freedom.

[Figure]

Figure R1: Frequency of in-situ observations at 5° longitude × 5° latitude bins in 2016.

[Figure]

Figure R2: *ΔNBE* averaged over the whole domain in the RTPP09+IAU experiment.

**■ Reviewer 2**

The manuscript discusses a setup of an ocean circulation model (the Stony Brook Parallel Ocean Model, sbPOM) for the north-western Pacific region combined with an LETKF data assimilation step. Daily assimilation of satellite and in situ observations is applied and sensitivity experiments are performed with and without incremental analysis updates (IAU) in which the parameters of different covariance inflation methods (in particular RTPP and RTPS) are varied. In addition, a multiplicative inflation is tested with a single fixed inflation value. The study finds that IAU improves the balance of the model increments while the inflation schemes disturb the balance. In contrast IAU leads to higher estimation errors and less ensemble spread than the inflation methods. The multiplicative inflation is found to be failing by not reducing error enough. Parameter ranges are described in which the different methods yield the best assimilation results (low imbalance combined with low estimation errors) and the overall conclusion if that IAU in combination with RTPP with a parameter value of 0.8-0.9 provides the best configuration.

Overall, I have large problems to find what is actually new in this study and what are relevant research results. Actually, while the authors write 'This study develops an ... (EnKF)-based regional ocean data assimilation system' (Abstract line 12), this system is certainly not new. Actually, Miyazawa et al. (2012) already described an LETKF in combination with the sbPOM model. This earlier publication did not use the same model configuration, but this implies that an actual LEKTF-sbPOM DA system already exists for 10 years and this leaves the impression that in the manuscript the authors (Y. Miyazawa is one of the co-authors) merely present some new model configuration. Even more, the applied methods IAU, RTPS and RTPP are established standard methods for ensemble data assimilation. Thus, it is unclear what new insight the experiments described in the manuscript actually provide. The given numbers like 'RTPP with the parameter of 0.8-0.9' (Abstract line 26) are not at all generalizable to other model configurations or other models. Further, the authors do not show any attempt to actually find explanations for their findings. As such it remains that they describe the behavior of a single data assimilation application when parameters of established standard methods are varied. For me, this is insufficient for a scientific publication. To this end, I can only recommend to reject the manuscript. Perhaps, the authors can then find a proper scientific question to assess with this ocean DA system and submit a new study that provides general insights.

We appreciate the reviewer for your comments. As indicated by the reviewer, Miyazawa et al. (2012) is the first paper to construct a regional data assimilation system, the sbPOM implemented with the LETKF, and performed experiments for a short period of about 1 month. It is because their system cannot provide realistic spatial patterns for temperature, salinity, and sea surface height if the experiment period extends over a few months, as added the description to the third paragraph in Section 1. This is similar to the results in the MULT and MULT+IAU experiments. Consequently, it is required to explore an appropriate setting for the sbPOM-LETKF system to represent accurate analysis fields.

We have substantially developed the sbPOM-LETKF system by implementing the perturbing atmospheric forcing method, IAU, RTPP, and RTPS, and have conducted the sensitivity experiments on the covariance inflation and IAU methods for a relatively long period of about 1.5 years. As a result, this study demonstrates that only the combination of the RTPP and IAU improves both accuracy and dynamical balance, and therefore it is the most suitable. To the best of our knowledge, only one among the IAU, MULT, and RTPS is adopted in existing ocean data assimilation systems (Table 1), and there are no studies to compare the impacts of covariance inflation and IAU methods on dynamical balance, accuracy, and ensemble spread. Therefore, we expect that this paper is helpful for readers to newly construct an EnKF-based ocean data assimilation system and improve the existing systems, although the suitable RTPP parameter would depend on tuning parameters and experimental setting. Since only the MULT is implemented with the LETKF source code on Github (https://github.com/takemasa-miyoshi/letkf), readers might choose MULT and face similar problems if readers do not find this paper.

Apart from the aspect of novelty and relevance, I have a few major comments:

1. The manuscript is submitted as a 'development and technical paper' and its title suggests that it might document particularities of the EnKF-sbPOM model system. However, the manuscript is missing detailed descriptions of the actual system.

The IAU, perturbed boundary conditions, RTPP, and RTPS are not incorporated into the system constructed by Miyazawa et al. (2012). This indicates that we have substantially developed the sbPOM-LETKF system. The detailed descriptions of the IAU, perturbed boundary conditions, RTPP, and RTPS have been included in Section 2, and the detailed setting of sbPOM and LETKF has been specified in Section 3 in the original and revised manuscripts.

2. The authors list EnKF-based ocean data assimilation systems in Table 1. Unfortunately, this list is very incomplete. E.g. there are EnKF/based system run operationally by the Copernicus Marine Service (CMEMS) for the global ocean and for the Baltic Sea (It is easy to find these systems via the CMEMS website marine.copernicus.eu). From the operational CMEMS systems, Table 1 only lists the TOPAZ4 system. There is also an operational EnKF-based system in Germany (the latest article about it is Bruening et al, 2021, but there are several publications about earlier versions dating back to the year 2012. This system uses 12-hourly analysis, thus even shorter than what is pointed out in the manuscript). Also there is an EnKF-based coupled system which focuses on the ocean (e.g. Tang et al. 2020). Overall the authors should perform a much more careful research on current systems. Publications dating back to 2011 or 2012 do most likely not describe the current status.

We thank the reviewer for letting us know about the ocean and coupled data assimilation systems. As indicated by the reviewer, there are ocean and coupled data assimilation systems not listed in Table 1. However, this study does not aim to summarize all of them, and therefore we mainly refer to ocean data assimilation systems related to this study. Nevertheless, we have missed a regional system for the North Sea and Baltic Sea constructed by Bruening et al. (2021), in which only the satellite SSTs are assimilated at short interval of 12 hours. We have added the system to Table 1 and the description to the third paragraph in Section 1. Although we have carefully searched EnKF-based global ocean data assimilation systems on the CMEMS Web site, we could not find such systems.

We have found several EnKF-based coupled data assimilation systems (Brune et al. 2015; Chang et al. 2013; Counillon et al. 2016) in addition to Tang et al. (2020), but they do not assimilate all typical observations (SST, SSH, T, and S) at a frequent interval similar to the existing EnKF-based ocean data assimilation systems. We have added a brief description of coupled data assimilation systems at the end of the third paragraph in Section 1.

3. The authors express that their data assimilation setup is particular because of daily assimilation. However, when one has a sufficiently complete overview one sees that short assimilation cycles like daily are not that special. On the other hand there are good reasons for longer cycles. One particular reason is the repeat cycle of the altimetry satellite data. Further, while applying e.g. weekly analyses steps, systems like TOPAZ4 use asynchronous filtering, e.g. for SST. Thus, the system is able to also take some of the faster variability into account. The authors should take such characteristics of the DA

systems into account to provide a sound overview of EnKF-based ocean DA systems.

Using a regional atmospheric data assimilation system, Maejima and Miyoshi (2020) demonstrated that the accuracy for 3D-LETKF is better than 4D-LETKF, which is similar to asynchronous filtering, although the computation cost of 3D-LETKF is higher. The satellites now provide the huge amount of SST observations at a frequent interval, although the daily distribution of satellite SSH is sparse. To maximize the use of the satellite SSTs, data assimilation at a frequent interval by 3D-LETKF would be better than the 4D-LETKF with a 1-week window.

4. As mentioned above, IAU, RTPP and RTPS are standard methods in DA already for quite some years. As such it is surprising to still see a manuscript submission about these schemes. Unfortunately, the authors also miss to take into account the study by Yan et al. (2014), which discusses IAU in ocean data assimilation. However, also the CMEMS system for the global ocean uses IAU. Given that these methods are well established and well studied, I am quite skeptical that it is possible to find new general insights by just using standard methods and varying their parameters.

Using an EnKF-based regional ocean data assimilation system, Yan et al. (2014) investigated the impacts of the IAU on dynamical balance and accuracy in twin experiments with an idealized setting, whereas this study conducts the sensitivity experiment assimilating real satellite and in-situ observations. Although a global data assimilation system in CMEMS (https://resources.marine.copernicus.eu/product-detail/GLOBAL_MULTIYEAR_PHY_001_030/INFORMATION) with singular evolutive extended Kalman filter (SEEK) adopts IAU, the system does not reveal the effects of the IAU on the dynamical balance and accuracy. The results for dynamical balance and accuracy in this study are consistent with Yan et al. (2014), and we have added the descriptions to the first paragraphs in subsections 4.1 and 4.2.1.

Although the IAU, MULT, RTPP, and RTPS are now well used in data assimilation field, to the best our knowledge, there are no studies to evaluate their impacts on dynamical balance, accuracy, and ensemble spread, combining the IAU and covariance inflation methods in an EnKF-based ocean data assimilation system. As clear from Tables 1 and 4, the combination of the RTPP and IAU is the most suitable but has not adopted in the existing EnKF-based systems. Therefore, this study would be helpful for readers to newly construct an EnKF-based ocean data assimilation system or improve the existing systems. However, as indicated by Reviewer #1, the appropriate RTPP parameter might

depend on other tuning parameters and experimental settings. We have added the description to the second paragraph in Section 6 in the revised manuscript.

5. The authors use a model spin-up of 4.5 years from an ocean in rest. This spin-up period looks far to short for properly spinning up the ocean unless one only looks at the upper layers.

If spatially uniform temperature and salinity are used for initial conditions, spinning up over several decades would be required. As described in subsection 3.1, however, observational monthly and seasonal temperature and salinity climatologies from the WOA18 are used for initial conditions with no motion, and the model is spun up with nudging temperature and salinity toward the monthly and seasonal climatologies with a 90-day timescale throughout the depth. Consequently, qualitatively similar results would be obtained regardless of the length of spin-up period. The long spin-up integration with 100 ensemble members is computationally expensive, and therefore we decide to choose a relatively short spin-up period.

6. The observation errors of 1.5degC for satellite SST and in situ temperature and of 0.2m for SSH are very large compared to what is commonly used today.

As seen in Ohishi et al. (in review), the low-salinity structure in the intermediate layer is degraded around the Kuroshio Extension if smaller temperature observation errors of 1.0 °C are applied. Consequently, temperature (SSH) observation errors are assumed to be 1.5 °C (0.2 m) in this study.

7. In lines 220-221 it is described that the localization settings are chosen following the studies by Miyazawa et al. (2021) and Penny et al., (2013). However, in these studies other model configurations with different resolutions are used and both use different localization radii. It is known that localization settings depend also on the model configuration. To this end, just selecting some settings from model configurations at other resolutions is not a reasonable approach. One can use values from other studies as a starting point for ones own tuning, but this tuning will be required as otherwise, there is a high risk that the DA system is suboptimal. Thus sub-optimality then also influences other DA parameters like those for the inflation.

As discussed in the 2nd paragraph in Sections 5 and 6 in the original and revised

manuscript, respectively, we have noticed that the localization scale is a tuning parameter and might depend on other tuning parameters such as covariance inflation parameters. It is beyond the scope of this study and an issue in future studies to survey an appropriate localization scale.

8. In line 60 the authors describe the TOPAZ4 system with 'but with inflation of observation errors'. I'm unsure what the authors intend to express by 'but'. However, when the authors look carefully, the 'moderation of observation errors' used in TOPAZ4 is in fact a careful inflation that should have similar effect as a carefully tuning multiplicative inflation scheme.

As described in the first paragraph in subsection 3.2 in Sakov et al. (2012), to prevent filter divergence, the TOPAZ4 multiplies observation errors by a factor of 2 when the ensemble perturbations are updated. Since the MULT parameter is generally set larger than one (i.e., $\rho > 1$), this procedure deflates the forecast ensemble spread and has opposite effects to the MULT. Furthermore, there are no descriptions for tuning the factor, and the observation error matrices for analysis ensemble mean and perturbation update should be consistent in the formulation of EnSRF and ETKF. Although adaptive observation pre-screening method to prevent an excessive shock is described in the second paragraph in subsection 3.2 in Sakov et al. (2012), this appears not to follow any theories such as the statistic innovation (Desroziers et al. 2005). Therefore, we could not find reasonable descriptions for the observation error inflations, and have maintained the description "*the TOPAZ4 uses all types of observations but with inflation of observation errors.*" in the third paragraph in Section 1.

9. The multiplicative inflation schemes is described as 'not demonstrate sufficient skill'. This description is actually misleading and invalid. The authors only run a single experiment with a fixed inflation of 5%. Thus, any sensitivity assessment is missing. Actually, the data assimilation process in the system of the manuscript runs already stable with successful assimilation even without inflation as the figures show. This is a clear indication that 5% multiplicative inflation is too large.

We thank the reviewer for pointing out the description that might mislead the readers. To avoid it, we have specified the parameter of the MULT in Abstract.

Following the comments from Reviewer #1, we have estimated the MULT parameter corresponding to the RTPP09+IAU experiment, and have added the results as

Section 5 in the revised manuscript. The spatiotemporal averaged estimated MULT parameter for SST, SSS, SSH fields are 1.08, 1.08, and 1.11, respectively, and correspond well to the prescribed MULT parameter ($\rho = 1.05^2 \approx 1.10$). Nevertheless, the results are completely different between the MULT+IAU and RTPP+IAU experiments.

Adaptive MULT may be helpful to estimate appropriate MULT parameter. As described in Ohishi et al. (in review), however, the AOEI improves the dynamical balance and accuracy of the temperature, salinity, and surface horizontal velocities. Since the AOEI has opposite effects to the adaptive MULT, the adaptive MULT would degrade the dynamical balance and accuracy.

It is difficult to explore the suitable MULT parameter, but we cannot deny that the suitable MULT parameter does not exist. Therefore, as described in the third paragraph in Section 5 (6) in the original (revised) manuscript, the MULT might have suitable parameter to improve the dynamical balance and accuracy.

10. The residual of the nonlinear balance equation {¥Delta}NBE (Eq. 8) is not normalized. As such it is unclear whether any of values shown in Fig. 1 and described in the text (like 2.11x10^{-10} for MULT+IAU in line 249) is actually significant.

In the first paragraph in subsection 4.1, spatiotemporal averaged $\Delta NBE$ is compared among the sensitivity experiments, and $\Delta NBE$=2.11$\times 10^{-10}$ and $5.22 \times 10^{-10}$ $s^{-2}$ in MULT and MULT+IAU experiments, respectively, is much larger than the other experiments. This study has shown the raw values of $\Delta NBE$ to directly compare among the sensitivity experiments.

References:

Bruening, T., Li, X, Schwichtenberg, F., Lorkowski, I. (2021) An operational, assimilative model system for hydrodynamic and biogeochemical applicatios for German coastal waters. Hydrographische Nachrichten, 118, 6-15, doi:10.23784/HN118-01

Tang, Q., Mu, L., Sidorenko, D., Goessling, H., Semmler, T., Nerger, L. (2020) Improving the ocean and atmosphere in a coupled oceanâ   atmosphere model by assimilating satellite sea surface temperature and subsurface profile data. Q. J. Royal Metorol. Soc., 146, 4014-4029 doi:10.1002/qj.3885

Yan, Y., Barth, A., Beckers, JM. (2014) Comparison of different assimilation schemes in

a sequential Kalman filter assimilation system, Oce. Mod. 73, 123-137, doi:10.1016/j.ocemod.2013.11.002

References:

Brune, S., Nerger, L. and Baehr, J.: Assimilation of oceanic observations in a global coupled Earth system model with the SEIK filter, Ocean Model., 96, 254–264, doi:10.1016/j.ocemod.2015.09.011, 2015.

Chang, Y. S., Zhang, S., Rosati, A., Delworth, T. L. and Stern, W. F.: An assessment of oceanic variability for 1960-2010 from the GFDL ensemble coupled data assimilation, Clim. Dyn., 40(3–4), 775–803, doi:10.1007/s00382-012-1412-2, 2013.

Counillon, F., Keenlyside, N., Bethke, I., Wang, Y., Billeau, S., Shen, M. L. and Bentsen, M.: Flow-dependent assimilation of sea surface temperature in isopycnal coordinates with the Norwegian Climate Prediction Model, Tellus, Ser. A Dyn. Meteorol. Oceanogr., 68(1), doi:10.3402/tellusa.v68.32437, 2016.

Maejima Y, Miyoshi T (2020) Impact of the window length of four-dimensional local ensemble transform Kalman filter: A Case of convective rain event. SOLA 16:37–42. https://doi.org/10.2151/sola.2020-007

Ohishi, S., Miyoshi, T., and Kachi, M.: An EnKF-based ocean data assimilation system improved by adaptive observation error inflation (AOEI), Geosci. Model Dev. Discuss. [preprint], https://doi.org/10.5194/gmd-2022-91, in review, 2022.

Recommendation: Major revision

Summary

This manuscript describes the local ensemble transform Kalman filter (LETKF) implemented in the Stony Brook Parallel Ocean Model (sbPOM), with daily assimilation of satellite and in-situ observations. Sensitivity experiments with IAU and various multiplicative inflation methods are performed. Results show that the application of IAU improves the analysis balance, but degrades the analysis accuracy and also reduces ensemble spread. The constant multiplicative inflation with or without IAU had much larger imbalances and errors than the other configurations. RTPP and RTPS with IAU had improved balances and smaller errors when the inflation parameter is tuned. The presentation of the manuscript is fine, and the lessons of inflation and IAU with influences on imbalance and accuracy are useful for the ocean DA community. But the results need further clarifications and explanations. Please see my comments below.

We thank the reviewer for constructive comments, especially on the IAU method.

1. It is confusing about the impact of IAU on the assimilation results. Compared to NOINFL, IAU in NOINFL+IAU degrades the accuracy. Why IAU degrade the accuracy for ocean assimilation that has longer time scale than atmosphere?

   The main difference between without and with the application of the IAU is directly updated the SSH or not. Temperature, salinity, horizontal velocities, and SSH analyses are used for the initial conditions for model integration within the assimilation window if the IAU method is not applied, whereas the analysis increments of temperature, salinity, horizontal velocities except for the SSH are distributed if the IAU method is applied. Therefore, the direct update of the SSH would result in higher accuracy of the SSH, SSHA, and surface horizontal velocities in the experiments without the IAU.

2. The authors state that IAU reduces the spread and accuracy of DA. But MULT, RTPP and RTPS have totally different impacts on the spread and accuracy when IAU is applied. Why MULT that also inflate the ensemble spread has the opposite impacts on spread and accuracy than RTPP and RTPS? Since the results with different inflation methods are inconsistent, it would be helpful to understand the roles of

different inflation methods, especially the interactions with IAU.

As indicated by Ohishi et al. (in review), the exceedingly large temperature and salinity increments result in the degradation of the temperature, salinity, and surface horizontal velocities, because they induce exceedingly strong vertical diffusion through weakening density stratification around the Kuroshio Extension region. Therefore, such large increments are not favorable for maintaining the stratification.

The RTPP and RTPS relax the analysis ensemble perturbations toward the forecast ensemble perturbations. This implies that the analysis increments in the RTPP and RTPS would be smaller than the MULT, and the above degradation process might be suppressed.

3. Previous studies of IAU (e.g., Lei and Whitaker 2016, He et al. 2020) showed that IAU has more advantages for variables that are more influenced by imbalances that variables that are less influenced by imbalances. However, results here are inconsistent with the previous findings. IAU improves the accuracy of wind field more than the accuracy of height field (Figures 3 and 4). Please provide explanations or insights for these counter-intuitive results.

Figures 3 and 4 indicate the degradation of the accuracy of the SSH and surface horizontal velocities by the IAU rather than the improvement. The degradation of the accuracy by the IAU is consistent with He et al. (2020) who demonstrated that the accuracy of most variables is worser in the 3D-IAU experiment than experiment without the IAU when the assimilation windows are short of 1 and 3 hours [See table 3 of He et al. (2020)]; Lei and Whitaker (2016) who indicated that the accuracy of temperature and wind speed is worser in the 3D-IAU experiment than the experiment without the IAU using NCEP GFS experiments with assimilation of real observations [See fig. 8 of Lei and Whitaker (2016)]; and Yan et al. (2014) who showed that the IAU degrades the accuracy in twin experiments using an EnKF-based ocean data assimilation system [See table 3 of Yan et al. (2014)].

4. Details of how the verification are done are needed. Which time is the imbalance deltaNBE computed at? Is it the prior or posterior at middle of DA window? The RMSD is computed for the prior or posterior? How the RMSD is computed for experiments with IAU?

Since $\Delta NBE$ can be calculated only at the assimilation timestep, it is calculated at

the beginning (middle) of the data assimilation window in the experiments without (with) the IAU. As described in the last paragraph in subsection 3.3 in the original and revised manuscripts, "We estimate $\Delta NBE$ from ensemble analysis increments on days 1 and 16 of each month, the RMSDs from the daily averaged ensemble mean, and the ensemble spread from the daily-mean ensemble."

5.  Since assimilation is conducted at a daily frequency, both the daily prior and free forecast at longer forecast lead times worth to check.

We thank the reviewer for constructive comments. To perform a free forecast after every assimilation cycle, all experiments must to be integrated again, and the huge amounts of the computational resources are required. Consequently, this is an issue in future studies.

Minor comments:
L90, for the IAU configuration here, is the analysis computed at the middle of an DA window or not? The 1.5 times computational cost is compared to the standard method with or without IAU? It is not clear why analysis is performed at the beginning of an DA window.

As described in subsection 2.1 in the original and revised manuscripts, the assimilation is conducted at the middle of window, and the computational costs with the IAU are 1.5 times those of the standard method (i.e., without the IAU).

The time lag between the forecast and observation becomes large if the assimilation is conducted at the beginning of an assimilation window, and therefore we have chosen the middle of the window for an assimilation timestep as proposed by Bloom et al. (1996).

References

He, L. Lei, J. S. Whitaker, and Z.-M. Tan, 2020: Impacts of Assimilation Frequency on Ensemble Kalman Filter Data Assimilation and Imbalances. J. Adv. Model. Earth Syst., 12, e2020MS002187.

Lei, L., and J. S. Whitaker, 2016: A four-dimensional incremental analysis update for the ensemble Kalman filter. Mon. Wea. Rev., 144, 2605-2621. doi:

http://dx.doi.org/10.1175/MWR-D-15-0246.1.

Yan, Y., Barth, A. and Beckers, J. M.: Comparison of different assimilation schemes in a sequential Kalman filter assimilation system, Ocean Model., 73, 123–137, doi:10.1016/j.ocemod.2013.11.002, 2014.

---

## Author Response (AR2)

Dear authors,

After reading the revised manuscript, I have to agree with critical comments of reviewers. Since the manuscript is mainly based on tuning parameters of established inflation methods RTPP and RTPS and their combinations with the IAU, from the scientific point of view, it already lacks innovation to some degree. Then it is hoped that authors could have made thorough discussion of results, unfortunately, it is also missing in the revised manuscript. I strongly recommend authors to improve the manuscript in three aspects as follows:

We thank the editor for reviewing our paper and providing useful comments. The dynamical balance is important for EnKF-based ocean data assimilation systems in which satellite and in-situ observations are assimilated at frequent interval. This study reveals that the combination of the IAU and RTPP only improves both dynamical balance and accuracy, by conducting the sensitivity experiments on various settings for IAU and covariance inflation methods. As indicated by Reviewer 2, the RTPP and IAU now might be standard methods, especially for the atmospheric community. However, to the best of our knowledge, most of EnKF-based ocean data assimilation systems have not adopted the IAU nor RTPP/RTPS (Table 1), and there are no studies to indicate that the combination of the IAU and RTPP is the best for dynamical balance and accuracy and to adopt it to EnKF-based ocean data assimilation systems.

Although this study does not include an object to quantitatively investigate the detailed mechanisms how the IAU and each covariance inflations affect the dynamical balance and accuracy, we reply as much as possible following the editor's comments.

1)Please make the research question more clear. As reviewers pointed out, the best combination could vary, for instance, if the ensemble size or localization change. Why do author think that tuning RTPP and RTPS in combination of the IAU is most worth exploring.

In the third paragraph in Section 1, we have discussed that dynamical imbalance might degrade the accuracy in frequent data assimilation. *To provide accurate analyses in EnKF-based ocean data assimilation system in which satellite and in-situ observations are assimilated at a frequent interval, it is necessary to investigate an optimal setting for both dynamical balance and accuracy.* We have added this description at the end of this paragraph.

In the fourth paragraph in Section 1, we have described that the IAU and covariance inflation methods might be important schemes for dynamical balance and accuracy. To indicate the importance of the choice the IAU and covariance inflation, we have added the description of "*In EnKF-based ocean data assimilation system, how to apply the analysis update to the model evolution and how to inflate the ensemble spread could make significant differences for the dynamical balance and accuracy. However, the IAU and RTPP/RTPS has not been widely used in an EnKF-based ocean data assimilation systems (Table 1).*" Thus, following the editor's comment, we have attempted to specify the research question: to investigate the optimal schemes for IAU and covariance inflation methods to obtain accurate and balanced analyses.

As described in the second paragraph in Section 6, the suitable setting, especially for the RTPP parameter, depends on the other tuning parameters and experimental settings. However, the results in this study would be helpful for readers who newly establish and develop EnKF-based ocean data assimilation systems.

2)It is disappointing that authors attempt to bypass reviewers' critical comments either by saying that it is out of the scope of this study or by that it is in line with published articles. By doing so, the study loses its own values and becomes irrelevant for the broader community which may have to deal with the same issue. In my opinion, two things here can be accomplished. 1) Currently, authors only present the results but quite few efforts have been made to explain them. The places worth explaining have been pointed out by reviewers. Please avoid to use the expressions like "out the scope" or "in line with...". The best tuned parameters can be different in different systems or in different settings of the same system, therefore, it is necessary to understand why those values turn out to be the best. 2) As known, imbalance has great impacts on the forecast skills, it is inappropriate to conclude which combination is the best without showing their forecast skills.

We have used "out of the scope" and "future studies" only when the huge amounts of computational resources are required to conduct additional experiments for long periods (Reviewer 1 Major comment #4 and Reviewer 3 Major comment #5) and to investigate tuning parameters such as localization scale (Reviewer 1 Major comment #3 and Reviewer 2 Major comment #7). To show the correspondence to previous studies following the fourth major comment from Reviewer 2, we have additionally cited only Yan et al. (2014) who investigated impacts of the IAU on vertical velocity and accuracy in twin experiments using a relatively idealized EnKF-based ocean data assimilation

system. We hope your understanding that we have attempted to reply to reviewers' and editor's comments to the best of our ability.

Regarding the first point, we have included the following descriptions to provide the insight into how the IAU and covariance inflation methods contribute to dynamical balance and accuracy. In the first paragraph in subsection 4.1, the IAU reduces the noise from high-frequency gravity waves associated with initial shocks and improves the balance. In the first paragraph in subsection 4.2.1, in contrast, the IAU degrades the accuracy because the ensemble spread is relatively small and the IAU does not use the SSH analysis increments. The small analysis increments might result in better dynamical balance. In the first paragraph in subsection 4.2.1, RTPP and RTPS improves accuracy by inflating the ensemble spread but the resulting large analysis increments might reduce dynamical balance. In Section 5, we have discussed why the MULT does not work well. We have also added the discussion why the combination of the IAU and RTPP with relaxation parameter of $\alpha_{RTPP} = 0.8 - 0.9$ is the most suitable at the end of the first paragraph in Section 6.

Regarding the second point, we have to conduct all experiments again from the beginning if we investigate the accuracy of the prediction similar to Lei and Whitaker (2016) and He et al. (2020) as replied to Reviewer 3, and currently we do not have the huge amounts of computational resources for this purpose. Daily-mean outputs in all sensitivity experiments rather than instantaneous analyses are used to evaluate the accuracy, and the $\Delta NBE$ and RMSDs are relatively stable throughout the experimental periods in all experiments except for the MULT and MULT+IAU experiments. Consequently, the prediction accuracy during the short period of a few days would be the qualitatively same as the accuracy estimated from the daily-mean outputs.

3)I have the same feeling as the reviewer, it is difficult to understand the symbols for statistical significance. I am sure there is a better way to present this.

Following the comments from Reviewer 1 and the editor, we have modified the symbols in Figs. 1, 3, 4, 7, and 8. Throughout the manuscript, black stars, open circles and triangles, and closed circles and triangles denote the NOINFL+IAU experiment, significant improvement and degradation, and improvement and degradation with no significant differences, respectively.

Last but the least, I am interested in the definition of imbalance. Does it include the ageostrophic component? To see clean impacts on imbalance caused by data assimilation,

is it better to calculate the imbalance difference between background and analysis?

$\Delta NBE$ defined in this study include only the geostrophic velocity except for the ageostrophic velocity. Here, we assume that the ageostrophic velocity is caused by the wind stress only except for geostrophic shear following the classical Ekman theory (Cronin and Tozuka 2016). The wind field is not modified by data assimilation in this system, and therefore the ageostrophic velocity would not be changed. Therefore, it is not necessary to include the ageostrophic velocity in the formulation of the non-linear balance equation (NBE).

$\boldsymbol{u}^a = \boldsymbol{u}^f + \delta\boldsymbol{u}$ and $\eta^a = \eta^f + \delta\eta$ is substituted into the geostrophic equation, where superscripts $a$ and $f$ indicate analysis and forecast, respectively. Since the geostrophic balance is satisfied in the forecast field, the increment terms only remain in Eq. (6). The solution is theoretically same if the analyses itself are used.

■ Reference

Cronin MF, Tozuka T (2016) Steady state ocean response to wind forcing in extratropical frontal regions. Sci Rep 6:28842. https://doi.org/10.1038/srep28842

---

## Author Response (AR3)

Before replying to the reviewer, we must inform that we found a bug in a code to calculate the RMSDs relative to the KEO buoy and then corrected it (Figs. 8 and 9 in the revised manuscript). Specifically, the number in the denominator is smaller in the RMSD calculation, and the corrected results show larger RMSDs for all experiments than the previous results. However, the corrected results are qualitatively the same as the previous results. Therefore, this correction has few impacts on the conclusion in this paper. We apologize for the above.

**Referee #1 (Dr. Yue Ying)**

Summary: I found the revised manuscript improved in clarity and overall quality. However, the authors have not fully addressed several major issues the reviewers raised, which still limits the scientific merit of the paper. I have reiterated the major issues below, note that to address them you don't necessarily need huge amount of computational resource for additional experiments. I would strongly encourage the authors think in terms of what the readers can learn from your results and put more effort in the interpretation of results.

We thank the reviewer for reviewing again our manuscript. We have conducted ensemble forecast experiments using the forecasts from the sensitivity experiments, and confirmed that the results from the forecast accuracy are qualitatively the same as the analysis accuracy.

1. My biggest concern is that only analysis ensemble is used in diagnosing RMSD for DA performance. Is your goal only to make the best ocean reanalysis? If you are implementing an ocean prediction system I will assume the forecast accuracy does matter and shall be considered in the diagnosis. Only a few additional forecasts (10-day forecast from every 1st of each month?) will provide some evidence that the IAU+RTPP outperforms the CTRL in terms of both balance and accuracy. If you absolutely don't have additional computational resources, then consider checking the RMSD for the priors (1day forecasts before DA, you should have them already). The bottom line is that some evidence shall be shown that improved accuracy/balance at analysis time will lead to some improvement in the forecast.

We thank the reviewer for your comments on the forecast accuracy. We performed 11-day ensemble forecast experiments initialized once a month in 2016 (i.e., total 12 cases) by the forecasts from the NO INFL experiments with/without IAU,

RTPP09 experiments with/without IAU, and RTPS09 experiments with/without IAU, and then calculated the daily averaged ensemble mean forecast RMSDs relative to the independent drifter buoys. The results are shown in Figs. 4a, b and 7 in the revised manuscript, indicating that the forecast RMSDs of surface horizontal velocities are qualitatively the same as the analysis RMSDs. We have added the related descriptions to the abstract, the second paragraph in subsection 3.3, the last paragraph in subsection 4.2.1, and the first paragraph in Section 6.

2. The way imbalance is diagnosed is still problematic: currently delta\_NBE is defined on the analysis increments delta\_u and delta\_eta (f-a), which means you assume the prior (f) fields are completely balanced and imbalance is only introduced by DA. Is this really the case? Does some imbalance still persiste after 1 day forecast, especially when ocean time scales are quite long? Could it be better to compute NBE separately for the prior and the analysis, and compare the two separately for different DA methods? If you can demonstrate that prior from the IAU+RTPP has better balance (lower NBE) and accuracy (lower RMSD) than CTRL, I will be convinced that IAU+RTPP is truly the best setup.

Although the reviewer asks to calculate  $\Delta NBE$  for the forecast field, this seems not to be reasonable because the forecasts are outputs from the model and follow its dynamical theory. The surface horizontal velocities can be approximately decomposed as the geostrophic and ageostrophic velocities. Here, we assume that the ageostrophic velocities result from the wind stress curl except for vertical geostrophic shear according to the classical Ekman theory (Cronin and Tozuka 2016). In this system, the atmospheric field is not analyzed, and therefore the prior and posterior ageostrophic field would not be changed. Therefore, the analysis increments of the SSH and surface horizontal velocities are better to satisfy the geostrophic balance. The geostrophic imbalance in the analysis increments is likely to be the source of the initial shocks. Little initial shock would occur if the analysis increments satisfied the geostrophic balance. We have added the above description to subsection 2.4.1.

These two issues are essential and I hope the authors will not bypass them and try to fully address them.

**Other comments:**

3. A comment on novelty of the paper: while IAU and RTPP are themselves well documented already, the combination of the two in a real prediction system is novel and

should be potentially useful for the DA community. But the interpretation is the key here, you need to go beyond just showing that IAU works and RTPP works as expected, and combining them just adds the merit. Since the balance and accuracy cannot be simultaneously satisfied, is it a matter of just finding the trade off? If I am making a reanalysis and don't care about prediction, can I just forget about imbalance (see that "best method" depends on application scenario)? Also, how can the readers be guided for their own implementation which method they shall use and how can they tune the parameters? Do you believe IAU+RTPP shall always perform well for all scenarios?

We thank the reviewer for your discussion on the novelty of this paper. The IAU is adopted in ocean data assimilation systems with *3D-VAR* and *4D-VAR* (See table 2 of Martin et al. 2015), the RTPP and RTPS are well used in the EnKF-based *atmospheric* data assimilation systems. However, as described in the fourth paragraph in Section 1. "the IAU and RTPP/RTPS have not been widely used in an EnKF-based ocean data assimilation systems (Table 1).", and unfortunately there are still few results how the IAU and RTPP/RTPS affect the dynamical balance and accuracy in EnKF-based ocean data assimilation systems. Although the most novelty part is the combination of the IAU and RTPP as indicated by the reviewer, this paper would have scientific significance for the ocean data assimilation community by comprehensively investigating the impacts of the IAU and covariance inflation methods on the dynamical balance and accuracy. The results are helpful for readers to newly construct or develop EnKF-based data assimilation systems in various fields including the ocean.

Based on the RMSDs of surface horizontal velocity in Fig. 4a, b, the combination of the RTPP and IAU is not a simple superposition. It is not easy to separate the effects of the RTPP and IAU in the RTPP+IAU experiments, probably because the RTPP and IAU appear to interact with each other in terms of the balance and accuracy. However, the results from the IAU, RTPP, and RTPP+IAU experiments imply that the RTPP maintains the ensemble spread inflated by the perturbed boundary conditions and results in the improvement of the accuracy but the degradation of the balance, and at the same time the IAU improves the degraded dynamical balance by the RTPP reducing the impacts from the initial shocks. This leads to further improvement of the accuracy. We have added the description at the end of the 1st paragraph in Section 6.

As described in the third paragraph in Section 1, the impacts from the initial shocks are likely to be accumulated and degrade the accuracy if frequent data assimilation is conducted, and a suitable setting to provide well-balanced analyses is necessary to construct accurate analysis products. Therefore, it is better to monitor the dynamical

balance if the assimilation interval is quite short.

Since the appropriate setting depends on systems as described in the second paragraph of Section 6, to find the best setting, it is necessary to comprehensively investigate the impacts of IAU and covariance inflation methods. Thus, we do not think that the RTPP+IAU is the best setting for all systems, but we expect that the RTPP+IAU improves the balance and accuracy in EnKF-based frequent data assimilation systems in which initial shocks have substantial impacts.

4. Temporal evolution and spatial distribution of the imbalance and inflation seem to me as the key to understanding the behavior (interplay between IAU and RTPP), you can try to pick a time series and visualize how NBE, RMSD, and inflation (a surrogate for spread reduction or where increment occurs in DA) fields evolve, instead of just showing their time- and domain-averages. The authors seem to consider this detailed analysis to be out of scope of the current study, but I think it is relevant. If you can provide some evidence that analysis accuracy and balance leads to better priors 1 day later it will convice us to use the IAU+RTPP setup.

Figure R1 shows spatially averaged  $\Delta NBE$ , analysis RMSDs of surface zonal velocity relative to the drifter buoys, and absolute SSH increment averages for each ensemble. Since  $\Delta NBE$  results from the geostrophic imbalance (the difference between the SSH gradient and surface horizontal velocity in the analysis increment fields), only the SSH and horizontal velocity increments cannot explain how  $\Delta NBE$  undergoes spatiotemporal variations. As shown in Fig. R1a, c, the timeseries of the SSH increment does not correspond to that of the  $\Delta NBE$ . Since the accumulated initial shocks result in the degradation of the accuracy in frequent data assimilation, it is reasonable that the timeseries of the RMSDs is not consistent with that of the  $\Delta NBE$  (Fig. R1a, b). Better balance and higher accuracy essential for frequent assimilation are well maintained in the RTPP09+IAU experiment than in the NO INFL experiment. We have already described the implication for interplay between the RTPP and IAU in the second paragraph in the reply to the third major comment. Furthermore, as replied to the first comment, we have demonstrated that the combination of the IAU and RTPP09 is the best for the forecast accuracy as in the analysis accuracy.

5. I fully understand the limit in computational resources, and in this case will not insist on tuning for the optimal parameters. The time evolution of averaged delta\_NBE indicates a clear seasonal cycle, so I would expect the best parameter will also have a seasonal cycle anyway. Maybe you can make a similar time series for the estimated inflation factor (you have shown the mean as 1.08, 1.11, and so on) to show that in practice an adaptive inflation/relaxation is better.

The best relaxation parameters would depend on time and space, and the adaptive RTPP and RTPS (Yue et al. 2015; Kotsuki et al. 2017) might be useful. This is a topic in future studies. Since the estimated MULT is used to investigate how much the inflation in the RTPP09+IAU experiment corresponds to the MULT parameter as described in the first sentence in Section 5, the estimated MULT does not enable us to estimate the suitable tuning parameter for the MULT and RTPP09+IAU from the estimated MULT.

**Reference:**

Cronin MF, Tozuka T (2016) Steady state ocean response to wind forcing in extratropical frontal regions. Sci Rep 6:28842. https://doi.org/10.1038/srep28842

**Referee #3**

I don't think the authors have addressed my previous comments. To be scientifically sound and a valuable contribution for the ocean DA community, the manuscript definitely need further and detailed explanations for the results.

We thank the reviewer for reviewing again our manuscript. We have confirmed the SSH increments and forecast accuracy as replied to the second and fourth comments, respectively, and also discussed the difficulty to decompose the interplay between the IAU and RTPP in the third comment.

1. To my previous comment 1, "Compared to NOINFL, IAU in NOINFL+IAU degrades the accuracy. Why IAU degrade the accuracy for ocean assimilation that has longer time scale than atmosphere?" The authors explain that "The main difference between without and with the application of the IAU is directly updated the SSH or not. Temperature, salinity, horizontal velocities, and SSH analyses are used for the initial conditions for model integration within the assimilation window if the IAU method is not applied, whereas the analysis increments of temperature, salinity, horizontal velocities except for the SSH are distributed if the IAU method is applied." I don't understand why different update variables are used for experiments with and without IAU. Please give the detail configurations of the experiments in the text, and also explain the reasons for the different choices of updated variables for different assimilation experiments.

We thank the reviewer for your comments on updated variables in the experiments with and without the IAU. In the experiments without the IAU (i.e., standard method), all analysis variables (SSH, temperature, salinity, and horizontal velocity) are used for initial conditions to keep the consistency. As seen in table 2 of Martin et al. 2015, there are three types of the IAU in ocean data assimilation systems using the following analysis variables:

- (i) temperature and salinity
- (ii) temperature, salinity, and horizontal velocity
- (iii) temperature, salinity, horizontal velocity, and SSH

All methods have advantages and disadvantages, and there are still discussions about which is better. We have chosen the second method in this study. It is because the SSH depends on the density, and SSH might be overcorrected if the analysis increments of the SSH, temperature, and salinity are applied to the IAU at the same time. We have added the above description at the end of subsection 2.1.

2. To my previous comment 2, "The authors state that IAU reduces the spread and accuracy of DA. But MULT, RTPP and RTPS have totally different impacts on the spread and accuracy when IAU is applied. Why MULT that also inflate the ensemble spread has the opposite impacts on spread and accuracy than RTPP and RTPS? Since the results with different inflation methods are inconsistent, it would be helpful to understand the roles of different inflation methods, especially the interactions with IAU." The authors referenced an under-review manuscript that is not available for the reviewers. And the explanation is "The RTPP and RTPS relax the analysis ensemble perturbations toward the forecast ensemble perturbations. This implies that the analysis increments in the RTPP and RTPS would be smaller than the MULT, and the above degradation process might be suppressed."

It would be helpful to see samples of analysis increments and subsequent forecasts, using MULT, RTPP and RTPS. However, the differences of analysis increments and subsequent forecasts with different inflation methods still cannot explain the interactions between IAU and inflation. Please provide understandings to the interactions of different inflation methods with IAU, which could help future design of data assimilation frameworks.

We apologized for not citing Ohishi et al. (in review) in the reply for Referee #3 and the revised manuscript, although it is cited in the previous reply comments for Referee #2. Ohishi et al. (in review) have been available through the GMD's website (https://gmd.copernicus.org/preprints/gmd-2022-91/gmd-2022-91.pdf). We have added the citation to reference in this reply comments for Referee #3 and the revised manuscript.

As shown in Figs. R1c and R2, the SSH analysis increments in the RTPP09 and RTPP09+IAU experiments are substantially smaller compared with the other experiments over most of the period. Here, we do not show the increments in the MULT and MULT+IAU experiments because they are exponentially increased and exceedingly large.

Although we have conducted sensitivity experiments on the IAU, covariance inflation methods, and their combination, their combination is not a simple superposition as shown by the analysis RMSDs of the surface horizontal velocity (Fig. 4a, b). Therefore, it is not easy to separate the effects of the IAU and RTPP in the RTPP+IAU experiments, probably because the IAU and RTPP interact with each other for balance and accuracy. This is true for understanding air-sea interaction. However, the IAU, RTPP, and RTPP+IAU experiments imply that the large relaxation parameter maintains the ensemble spread induced by perturbed boundary conditions and leads to the improvement of accuracy but the degradation of dynamical balance, and at the same time the IAU

improves the degradation of the dynamical balance by the RTPP. As a result, this would lead to further improvement of the accuracy by reducing the initial shocks in frequent data assimilation. We have added the description at the end of the first paragraph in Section 6.

3. To my previous comment 3, "Previous studies of IAU (e.g., Lei and Whitaker 2016, He et al. 2020) showed that IAU has more advantages for variables that are more influenced by imbalances that variables that are less influenced by imbalances. However, results here are inconsistent with the previous findings. IAU improves the accuracy of wind field more than the accuracy of height field (Figures 3 and 4). Please provide explanations or insights for these counter-intuitive results." The authors replied "The degradation of the accuracy by the IAU is consistent with He et al. (2020) who demonstrated that the accuracy of most variables is worser in the 3D-IAU experiment than experiment without the IAU when the assimilation windows are short of 1 and 3 hours [See table 3 of He et al. (2020)]; Lei and Whitaker (2016) who indicated that the accuracy of temperature and wind speed is worser in the 3D-IAU experiment than the experiment without the IAU using NCEP GFS experiments with assimilation of real observations [See fig. 8 of Lei and Whitaker (2016)]; and Yan et al. (2014) who showed that the IAU degrades the accuracy in twin experiments using an EnKF-based ocean data assimilation system [See table 3 of Yan et al. (2014)]."

First, Table 3 of He et al. (2020) showed that for the surface height that is more sensitive to imbalances than the other variables, 3DIAU is better than NoIAU for DA frequencies of 12h, 6h and 3h, while 3DIAU is worse than NoIAU for DA frequency of 1h. I don't think the 1h DA frequency can be extrapolated here, since here less frequent observations are assimilated, and the oceanic model has much longer time scales than the QG model. Second, He et al. (2020) showed that IAU can impact the surface height more than the wind, since the latter is less sensitive to imbalances. But in this study, IAU improves the accuracy of wind field more than the accuracy of height field (Figures 3 and 4). Please provide dynamical explanations for this result.

To focus on the points to be discussed, we only compare the results from the assimilation experiments at the finest interval of 1 hour in table 3 of He et al. (2020) with this study, because 1-day assimilation is a quite frequent interval for the ocean data assimilation systems. As seen in table 3 of He et al. (2020), the 3DIAU results in *larger* RMSEs and *degrades* the accuracy of all atmospheric variables (upper- and lower-layer

wind, interface height, and surface height). However, the reviewer incorrectly described "IAU *improves* the accuracy of wind field more than the accuracy of height field" in the previous and present comments, although we have indicated this in the previous comment. Therefore, the IAU *degrades* the accuracy of the SSH and surface horizontal velocity, and this is qualitatively the same as He et al. (2020). We note that He et al. (2020) conducted assimilation experiments at a 1-hour interval using hourly observations, and therefore their results are not extrapolated.

In the second point, we again note that the IAU *degrades* both SSH and surface velocity fields. As described in subsection 2.4.3, the SSH/SSHA from the AVISO and the surface horizontal velocity from the drifter buoys are completely different. Since the AVISO is constructed by summing optimally interpolated satellite SSHA and mean dynamical ocean topography estimated from the atmospheric datasets and drifter buoys, the AVISO is not independent dataset. In contrast, the surface drifter buoys are independent observations. Therefore, we cannot quantitatively nor directly compare the accuracy between the SSH and surface horizontal velocity. If forecast/analysis errors are accurately estimated by conducting twin experiments, the direct comparison is possible.

4. To the last question of my previous comment 4, "The RMSD is computed for the prior or posterior? How the RMSD is computed for experiments with IAU?" and my previous comment 5, "Since assimilation is conducted at a daily frequency, both the daily prior and free forecast at longer forecast lead times worth to check." The authors replied "To perform a free forecast after every assimilation cycle, all experiments must to be integrated again, and the huge amounts of the computational resources are required. Consequently, this is an issue in future studies." I totally understand the computational cost. But since cycling assimilation experiments are already done, it should be straightforward to calculate the verifications with priors, since no additional computations needed. Moreover, just several samples of long free forecasts from different assimilation experiments could be useful to draw some conclusions.

Because of the limitation of the storage, we have conserved only instantaneous forecasts and analyses (restart files) at the 1st and 16th days for each month, and daily averaged ensemble mean and spread throughout the whole period. Therefore, we have to conduct all experiments from the beginning if daily forecast RMSDs are calculated throughout the analysis period. Instead, we performed 11-day ensemble forecast experiments for each month in 2016 following the first major comment from Referee #1 (Fig. 7 in the revised manuscript). The results show that the forecast RMSDs are

qualitatively the same as the analysis RMSDs, and that the forecast accuracy is the best in the RTPP09+IAU experiment. Therefore, the combination of the IAU and RTPP is the most suitable for not only constructing the analysis products but also conducting ensemble forecast.

**Reference:**

Ohishi, S., Miyoshi, T., and Kachi, M.: An EnKF-based ocean data assimilation system improved by adaptive observation error inflation (AOEI), Geosci. Model Dev. Discuss. [preprint], https://doi.org/10.5194/gmd-2022-91, in review, 2022.

Figure R1: Spatial averaged (a)  $\Delta NBE$ , (b) analysis RMSDs of the surface zonal velocity relative to the drifter buoys, and (c) absolute SSH increments over the whole domain in the NO INFL (black), RTPP09 (red), RTPS09 (blue), NO INFL+IAU (gray), RTPP09+IAU (orange), and RTPS09+IAU (orange) experiments.

---

## Author Response (AR4)

**Referee #1 (Dr. Yue Ying)**

Thank the authors for careful revision of the manuscript and addressing the issues. The replies have been satisfactory and I would suggest the manuscript be accepted. Some very minor wording issues can be fixed, which I list below.

We thank the reviewer for carefully checking the manuscript. We have modified the manuscript following your comments.

**1) Line 105: ...because SSH depends on..., I suggest starting a new sentence here, and "might be" sounds uncertain, I suggest you say it more definitively like "SSH tends to be overcorrected if...".**

We have divided the one sentence into two sentences, and have replaced "might" with "tends to".

**2) Eq. 1: try to use [()] when parentheses are nesting (it is clearer).**

We have modified the outer parentheses in Eq. (1).

**3) Line 152: surface horizontal velocity can be approximately represented..., why "approximately", shouldn't it be "exactly" geostrophic + ageostrophic velocities? Or you define total velocity = geostrophic + ageostrophic + residual? Be precise here.**

Given the quasi-geostrophic velocity equation, it is obvious that the velocity should be decomposed into geostrophic, ageostrophic, and residual. In the first sentence in subsection 2.4.1, we have removed "approximately" indicating the assumption of the geostrophic approximation.

**4) Line 153: Instead of saying "ageostrophic velocity is assumed to be caused by...", you shall just state that in this study you "define" ageostrophic velocity as... (then I guess you have residual velocity to close the equation?)**

We have replaced "assumed" with "defined" in the second sentence in subsection 2.4.1. In the classical Ekman theory, the ageostrophic velocity is derived on the assumption that the geostrophic vertical shear is neglected. Cronin and Tozuka (2016) proposed the frontal Ekman theory where the geostrophic vertical shear has substantial impacts on the

ageostrophic velocity especially around the frontal regions.

**5) Line 155: "atmospheric field is not analyzed": Just mention that the surface wind stress (atmospheric field) is from the lateral boundary condition that is not part of the ocean model state vector (in the DA analysis).**

We have modified the third sentence in subsection 2.4.1 to indicate that the atmospheric field is not included in the model state vector.

**6) Line 175: Be precise in statements: If analysis increment satifies the geostrophic balance (or the NBE actually?), there will be "zero" Delta_NBE and "no" shock.**

As indicating the reply to the third comment, there might be residual velocity even if the geostrophic balance is completely satisfied in the analysis field. Therefore, we have maintained the last sentence in subsection 2.4.1 to give an accurate explanation.

**7) Line 185: Suggest rephrasing: "Significance of the improvement/degradation to the dynamical balance and analysis accuracy is tested in a bootstrap approach. We resample 10,000 cycles from the assimilation experiments and IRs at a 99% confidence level are considered significant"**

We have modified the last sentence in subsection 2.4.2 referring to the reviewer's comment.

**8) Line 309: "might result in a better", delete "might"**

We have removed "might" in the first paragraph in subsection 4.2.1.

**Referee #3**

Thank the authors for their efforts to address previous comments. In my opinion, the manuscript is significantly improved, especially with the forecast results.

We thank the reviewer for your helpful comments. We have added the forecast RMSDs of the SSH and SSHA relative to the AVISO to Fig.7 and confirmed that the results of the forecast RMSDs are qualitatively almost consistent with those of the analysis RMSDs.

1. I want to confirm that experiments with IAU (e.g., NO INFL + IAU, RTPP+IAU, RTPS+IAU) do not update SSH? At l105, "because SSH depends on density and might be overcorrected if the temperature, salinity, and SSH increments are used at the same time". I am curious that is there an IAU experiment tried with SSH increment? SSH could be more sensitive to imbalance than the other variables, and thus IAU might have stronger impact on SSH than the other variables. Since the sensitivity experiments without IAU updating SSH but sensitivity experiments with IAU not, there is a gap for explaining the results. Are the differences among the experiments with and without IAU from IAU only or from IAU and no SSH update? It would be helpful if the authors can shed insights on this gap between the two groups of sensitivity experiments.

As indicated by the original manuscript and our reply to the previous comment, the SSH increments are not used in all IAU experiments since the SSH increments tend to cause initial shocks. In the IAU experiment, the SSH is modified properly in response to the temperature and salinity increments. Table 2 of Martin et al. (2015) shows that no SSH update in the IAU is adopted in four out of six existing ocean DA systems, and therefore, it is considered as a common approach to ocean DA. We have added the related description to the end of subsection 2.1.

2. Thank the authors to run additional forecasts. Figure 7 shows the forecast errors at 11-d lead times for surface winds. How about the forecast errors for SSH and SSHA? It would also be helpful to show the errors at different forecast lead times.

Following the comments, we have added forecast RMSDs of the SSH and SSHA relative to the AVISO to Fig. 7 in the revised manuscript. The results shown in Fig. 7 generally agree with the results of the analysis RMSDs in Fig. 3, except for the RTPP09+IAU and RTPS09+IAU experiments showing improved forecast SSHA accuracy relative to the NO INFL+IAU experiment. We have added the description to the last paragraph in subsection

4.2.1. We have also confirmed that the results are qualitatively the same if the forecast period is changed.

---

## Author Response (AR5)

thanks for your efforts to improve the quality of the manuscript. From my point of view, I have some concerns. 1) I suggest that results of forecasts should be more rigorously discussed. As already done in the recent revision, results have been verified against the AVISO and the drifter buoys, therefore, it is natural to show results for verification against KEO buoy as well. Otherwise, it is not only less content but also abrupt in the context.

We have added the results of the forecast RMSDs relative to the KEO buoy to Fig. 9 in the revised manuscript. The results from the forecast RMSDs are qualitatively the same as those from the analysis RMSDs. We have added the related description to the first and last paragraph in subsection 4.2.2.

2) In Fig. 7, the RMSD for SSH of RTPP09+IAU is considerably larger, is there any explanation for this?

As consistent with the forecast SSH RMSDs, the analysis SSH RMSD in the RTPP09+IAU experiment is larger than the NO INFL experiment. The differences between SSH and SSHA RMSDs are caused by the mean dynamical ocean topography (MDOT) between the AVISO and data assimilation system. Here, the MDOT of the AVISO is estimated from a geoid model, satellite altimetry, and in-situ drifter buoy data, whereas that of the system is assumed to be the simulated SSH averaged in 2012–14. We have added the description to subsection 2.4.3 and the first paragraph in subsection 4.2.1.

3) As shown by other work (e.g., Zeng et al 2018, JAMES and Bowler et al. 2017, QJRMS), the RTTP tends to result over-balanced or too smooth analyzed fields, I wonder why the IAU still exhibits considerable advantage while applying the RTPP. Intuitively, I would have thought that the RTPS combined with the IAU should be more beneficial. I hope that authors could have some discussion on this.

The ocean acts as "memory" and tends to conserve effects from the initial shocks, and therefore ocean data assimilation systems are sensitive to initial shocks. In this system, the assimilation interval is short of 1 day, and therefore it is better to suppress the initial shocks as much as possible.
As consistent with the discussion in Whitaker and Hamill (2012), the results in this study show that the analyses are more balanced in the RTPP experiment than in the RTPS experiment (Fig. 1). More balanced analysis increments result in smaller initial shocks, and consequently the RTPP+IAU experiment would have better accuracy than the

RTPS+IAU experiment. We have added the description to the second paragraph in Section 6.

Zeng, Y., Janjić, T., de Lozar, A., Blahak, U., Reich, H., Keil, C., Seifert, A., 2018. Representation of model error in convective-scale data assimilation: additive noise, relaxation methods and combinations. J. Adv. Model. Earth Syst. 10, 2889–2911.

Bowler, N. E., A. M. Clayton, M. Jardak, E. Lee, A. C. Lorenc, C. Piccolo, S. R. Pring, M. A. Wlasak, D. M. Barker, G. W. Inverarity, and R. Swinbank (2017), Inflation and localization tests in the development of an ensemble of 4D-ensemble variational assimilations, Q. J. R. Meteorol. Soc., 143, 1280-1302.